# Sparse connectivity for MAP inference in linear models using sister mitral cells

**Sina Tootoonian**[1,2]*, **Andreas T. Schaefer**[2,3], **Peter E. Latham**[1]

**1** Gatsby Computational Neuroscience Unit, University College London, London, United Kingdom, **2** Sensory Circuits and Neurotechnology Laboratory, The Francis Crick Institute, London, United Kingdom, **3** Department of Neuroscience, Physiology & Pharmacology, University College London, London, UK

* sina.tootoonian@crick.ac.uk

**Data Availability Statement:** All code to generate the data and figures in this publication is publicly available at https://github.com/stootoon/sister-mcs-release.

## Abstract

Sensory processing is hard because the variables of interest are encoded in spike trains in a relatively complex way. A major goal in studies of sensory processing is to understand how the brain extracts those variables. Here we revisit a common encoding model in which variables are encoded linearly. Although there are typically more variables than neurons, this problem is still solvable because only a small number of variables appear at any one time (sparse prior). However, previous solutions require all-to-all connectivity, inconsistent with the sparse connectivity seen in the brain. Here we propose an algorithm that provably reaches the MAP (maximum *a posteriori*) inference solution, but does so using sparse connectivity. Our algorithm is inspired by the circuit of the mouse olfactory bulb, but our approach is general enough to apply to other modalities. In addition, it should be possible to extend it to nonlinear encoding models.

## Author summary

Sensory systems must infer latent variables from noisy and ambiguous input. MAP inference—choosing the most likely value for the latent variables given the sensory input—is one of the simplest methods for doing that, but its neural implementation often requires all-to-all connectivity between the neurons involved. In common sensory contexts this can require a single neuron to connect to hundreds of thousands of others, which is biologically implausible. In this work we take inspiration from the 'sister' mitral cells of the olfactory system—groups of neurons associated with the same input channel—to derive a method for performing MAP inference using sparse connectivity. We do so by assigning sister cells to random subsets of the latent variables and using additional cells to ensure that sisters correctly share information. We then derive the circuitry and dynamics required for the sister cells to compute the original MAP inference solution. Our work yields a biologically plausible circuit that provably solves the MAP inference problem and provides experimentally testable predictions. While inspired by the olfactory system, our method is quite general, and is likely to apply to other sensory modalities.

**Funding:** This work was supported by the Gatsby Charitable Foundation (P.E.L., https://www.gatsby.org.uk/), Wellcome Trust (https://wellcome.org/) Investigator grants 110174/Z/15/Z (A.T.S.), 110114/Z/15/Z (P.E.L.), and by the Francis Crick Institute (https://www.crick.ac.uk/), which receives its core funding from Cancer Research UK (FC001153), the UK Medical Research Council (FC001153), and the Wellcome Trust (FC001153). The funders had no role in study design, data collection and analysis, decision to publish, or preparation of the manuscript.

**Competing interests:** The authors have declared that no competing interests exist.

# 1 Introduction

A common view of sensory systems is that they invert generative models of the environment to infer the causes underlying sensory input. Sensory input is typically ambiguous, so a given input can be explained by multiple causes. Consequently, correct inference requires adequately accounting for interactions among causes. For example, increased evidence for one cause often reduces the probability of, or "explains away", competing causes (if you think the object you're smelling is an orange, that makes it less likely to be a lemon). Any neural circuit performing inference must therefore implement mechanisms for inter-causal interaction. This typically results in dense—and in many cases all-to-all—connectivity between neurons representing causes. The myriad causes potentially responsible for a given sensory input often require a neuron representing a cause to connect to hundreds of thousands of others. Such dense connectivity is biologically implausible.

This problem is easy to demonstrate in linear models of sensory input. (Although linear may seem overly restrictive, in fact such models have been successful in explaining basic features of the visual [1], olfactory [2], and auditory [3] systems). Consider noisy receptors $y_i$ (e.g. retinal ganglion cells, olfactory glomeruli) linearly excited by causes $x_j^*$ (e.g. edges, odours) according to a matrix $A_{ij}$. Under this model, the excitation of the $i$th receptor is given by

$$y_i = \sum_{j=1}^{N} A_{ij} x_j^* + \text{noise}. \tag{1}$$

The causes responsible for the observations, $y_i$, can be estimated by minimizing, for example, the squared error between the *actual* observations and the *expected* observations. A population of neurons whose individual firing rates represent the $x_j$ can do this by gradient descent [4],

$$\frac{dx_j}{dt} \propto -\frac{\partial}{\partial x_j} \sum_{i=1}^{M} \left( y_i - \sum_{k=1}^{N} A_{ik} x_k \right)^2 \propto \sum_{i=1}^{M} A_{ij} y_i - \sum_{k=1}^{N} \left( \sum_{i=1}^{M} A_{ij} A_{ik} \right) x_k. \tag{2}$$

These dynamics can be interpreted as balancing the evidence for the cause $x_j$ due to the receptor inputs $y_i$ (the first term) while accounting for the explanatory power of the other causes (the second term). In particular, $\sum_i A_{ij} A_{ik}$ reflects the contribution of cause $x_k$ to the evidence for cause $x_j$. Importantly, even if the elements $A_{ij}$ are sparse, $\sum_i A_{ij} A_{ik}$ will be non-zero for most $j$ and $k$, implying nearly all-all connectivity in a circuit implementing Eq (2). In common sensory settings there may be hundreds of thousands of causes that explain a given input. This means that $x_j$ must connect to hundreds of thousands of other neurons.

Below we show how the problem of all-to-all connectivity can be solved so that inference can be performed with realistically sparse connectivity. We begin by recapitulating the MAP inference problem, focusing on the olfactory setting for concreteness. This is basically sparse coding applied to olfaction, and suffers from all-to-all connectivity. We then derive a solution inspired by the anatomy of the vertebrate olfactory bulb, namely the presence of dozens of 'sister cells' that receive input from the same glomerulus. That solution leads to MAP inference, but using sparser connectivity. While we focus here on the olfactory system, our method is applicable to other modalities.

# 2 Results

## 2.1 Olfaction as MAP inference

Animals observe odours indirectly via the excitation of olfactory receptor neurons that project their axons to spherical bundles of neuropil called glomeruli. Each receptor neuron is thought

to express a single receptor gene from a large repertoire [5], and neurons expressing the same gene almost always converge onto one of two glomeruli, on either side of the olfactory bulb [6]. Thus each glomerulus represents the pooled activity of the receptor neurons expressing a single type of olfactory receptor. We represent this vector of glomerular activations by $\mathbf{y} = \{y_1, y_2, \ldots, y_M\}$, where $y_i$ is the activation of the $i^{\text{th}}$ glomerulus, and $M$ is the number of glomeruli per lobe of the olfactory bulb, or equivalently, the number of olfactory receptor genes expressed by the animal. This number is $\sim 50$ for flies [7], $\sim 300$ for humans [8], and $\sim 1000$ for mice [9].

The task of the animal is to infer the odour, $\mathbf{x}^*$ (which consists of $N$ components, $\{x_1^*, x_2^*, \ldots, x_N^*\}$), from the receptor activations, $\mathbf{y}$ (see Fig 1A). There are two main interpretations for the $x_j^*$. One is that $x_j^*$ is the concentration of the $j^{\text{th}}$ molecule in the odour, and so $N$ is the number of distinct molecular species that the animal may encounter in its environment. The other is that $x_j^*$ represents a complete olfactory object (e.g., coffee, bacon, marmalade) rather than a molecular species; in this case, $N$ is the number of learned odours. To estimate $N$ for the first interpretation, we note that the study of an estimated 0.25% of all flowering plants has yielded 1700 floral scent compounds [10], suggesting an upper estimate for $N$ on the order of 1700/0.0025, or roughly 700,000 (though the actual number could far fewer if existing molecules appear in as-yet-undiscovered floral scents), on the same order as the 400,000 estimated in the literature [11]. For the second interpretation (odours are complex olfactory objects), $N$ is difficult to approximate, but estimates for the number of *distinguishable* odour objects range from 10,000 [12] to 1 trillion [13]. Here we simply assume that in both cases $N$ is large. In

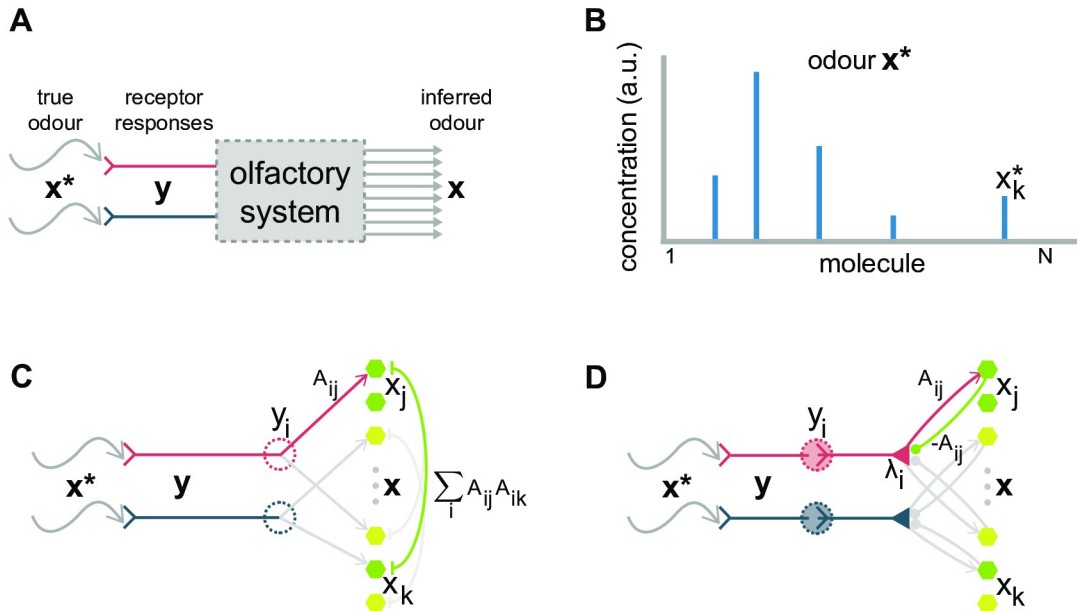

**Fig 1. Olfaction as MAP inference.** (A) Animals observe odours $\mathbf{x}^*$ indirectly via receptor activations. We assume the function of the olfactory system is to report the odour $\mathbf{x}$ most likely to have caused the observed receptor activations $\mathbf{y}$. (B) Schematic representation of an odour, whose defining feature is that it's sparse (meaning very few components are active). (C) A basic circuit for performing MAP inference on odours: Receptor $i$ projects directly to each readout unit $x_j$ with weight $A_{ij}$ determined by the affinity of receptor $i$ for molecule $j$; the readout unit $x_j$ reciprocally inhibits unit $x_k$ with weight $\sum_i A_{ij} A_{ik}$. The latter term is likely be non-zero even if $A_{ij}$ is sparse (since that requires only one term in the sum over $i$ to be nonzero), resulting in each readout unit inhibiting and being inhibited by potentially $\sim 100,000$ other units. (D) An alternate circuit that performs the same inference. Mitral cells now mediate between glomeruli and the readout units. Each mitral cell $\lambda_i$ excites each readout unit $x_j$ with weight $A_{ij}$ and is in turn inhibited by the same amount. No inhibition is needed between readout units, but a mitral cell must still excite and be inhibited by each of potentially $\sim 100,000$ readout units.

either case, odours are very sparse—either because only a few molecular species are present [14], or only a few odours are present.

From the point of view of learning and inference, there are advantages to both interpretations. If odours are represented as molecular concentration vectors, then information about how each molecule excites each receptor is determined only by the physical parameters of the molecule and receptor. It can, therefore, be learned on evolutionary timescales and hard-wired into the circuit, at least in principle, and it provides a simple substrate for the animal to generalize between chemically similar odours. It is disadvantageous in that the animal requires higher-order circuitry to infer learned olfactory objects (coffee, bacon, marmalade), which consist of many types of odour molecules. If odours are represented as complex objects, then those objects have to be learned within the lifetime of the animal. However, once learned, further higher order circuitry is not needed. Although these important representational issues are beyond the scope of this work, we mention them in passing as examples of the non-trivial assumptions required before a complete theory of olfactory circuit function can be developed.

We assume a very simple model of the transduction of odours into neural activations: odour components contribute linearly to the input current of a receptor, which is then converted into a firing rate by a static, invertible, pointwise nonlinearity. That is, the excitation of glomerulus $i$ is described as

$$y_i = f\left(\sum_{j=1}^{N} A_{ij} x_j^* + z_i\right), \quad z_i \sim \mathcal{N}(0, \sigma^2) \tag{3}$$

where $x_j^*$ is the concentration of the $j$th molecule, $A_{ij}$ is the *affinity* of the $i$th receptor for the $j$th molecule, $f$ converts input current to firing rate, and $z_i$ is additive noise with variance $\sigma^2$. Nonlinearities like $f$ can be inverted without changing the nature of the inference problem, so, without loss of generality, we take $f$ to be the identity. Thus, our likelihood is

$$p(y_i|\mathbf{x}) \sim \mathcal{N}\left(\sum_{j=1}^{N} A_{ij} x_j, \sigma^2\right). \tag{4}$$

Because the number of glomeruli, $M$, is likely to be much smaller than the number of molecular species, $N$, a whole manifold of odours can be consistent with any particular pattern of glomerular activation. We resolve this ambiguity by selecting the candidate odour that is most consistent with our prior information about odours. The prior $p(\mathbf{x})$ encodes the animal's background knowledge about the presence of odours in the environment. We assume that the animal makes the simplifying assumptions that molecules appear independently of each other (but see [15, 16]), and that the marginal probability distribution for each molecule, $p(x_i)$, has the same form.

For the prior we use an elastic net distribution (a combination of $\ell_1$ and $\ell_2$ penalties) [17, 18]. The $\ell_1$ penalty promotes sparsity as is observed [14] (see Fig 1B); the $\ell_2$ penalty discourages very large concentrations. In addition, we include a term enforcing the non-negativity of concentrations, yielding

$$p(\mathbf{x}) = \prod_{i=1}^{N} p(x_i) \propto \exp\left(-\sum_{i=1}^{N} \phi(x_i)\right) \tag{5}$$

where

$$\phi(x_i) = \beta|x_i| + \frac{\gamma}{2} x_i^2 + \mathbb{I}(x_i \geq 0). \tag{6}$$

The parameter $\beta$ determines the degree of sparsity, $\gamma$ penalizes excessively large

concentrations, and the indicator function, defined as $\mathbb{I}(x_i \geq 0) = 0$ when $x_i \geq 0$ and $\infty$ otherwise, enforces the non-negativity of concentrations.

The optimization problem is to determine the odour $\mathbf{x}$ most likely to have caused glomerular activations $\mathbf{y}$, taking into account both the likelihood and prior. The resulting MAP estimate is given by

$$\mathbf{x}_{\text{MAP}} = \underset{\mathbf{x} \in \mathbb{R}^N}{\text{argmin}} \sum_{j=1}^{N} \phi(x_j) + \frac{1}{2\sigma^2} \sum_{i=1}^{M} \left( y_i - \sum_{j=1}^{N} A_{ij} x_j \right)^2. \tag{7}$$

Because the objective function being minimized is strictly convex it has a unique minimum. At this minimum the partial derivative of the objective with respect to each $x_j$ (ignoring for the moment the potential non-differentiabilities introduced in Eq (6)) is zero, yielding

$$\frac{\partial \phi}{\partial x_j} = \frac{1}{\sigma^2} \sum_i A_{ij} \left( y_i - \sum_k A_{ik} x_k \right). \tag{8}$$

A common approach for solving this equation is to perform gradient descent on the objective function (the right hand side of Eq (7)) [1], for which the resulting dynamics is

$$\tau_x \frac{dx_j}{dt} = -\frac{\partial \phi}{\partial x_j} + \frac{1}{\sigma^2} \sum_i A_{ij} \left( y_i - \sum_k A_{ik} x_k \right). \tag{9}$$

(Note that we recover Eq (2) if we set $\beta$ and $\gamma$ to zero and $\sigma^2$ to 1, and drop the non-negativity constraint in Eq (6)). These dynamics have a natural interpretation: there's a leak term due to the gradient of the prior (the first term on the right hand side), feed-forward excitation of the readout unit $x_j$ by the glomeruli (the second term), and recurrent inhibition among the readout units (the third term); see Fig 1C.

As discussed above, the problem with this approach is that unless the affinity matrix $A_{ij}$ has sparse structure, the term $\sum_i A_{ij} A_{ik}$ leads to dense connectivity. To remedy that, we can factor out the term in parentheses in Eq (9) and implement it with a new variable, $\lambda_i$, giving us

$$\tau_\lambda \frac{d\lambda_i}{dt} = -\lambda_i + \frac{1}{\sigma^2} \left( y_i - \sum_{j=1}^{N} A_{ij} x_j \right) \tag{10a}$$

$$\tau_x \frac{dx_j}{dt} = -\frac{\partial \phi}{\partial x_j} + \sum_{i=1}^{M} A_{ij} \lambda_i. \tag{10b}$$

Although this describes a different neural system than that in Eq (9), it clearly has the same fixed point.

The circuitry implied by Eq (10) is broadly consistent with that of the olfactory system (see Fig 1D). There is one $\lambda_i$ for each glomerular input $y_i$, making it natural to identify $\lambda_i$ with the activation of a mitral or tufted cell. Mitral and tufted cells likely play different roles in olfactory processing [19], but our theory can be applied to both populations, so, for simplicity, we will refer to them collectively as mitral cells.

Eq (10a) requires each mitral cell to be directly excited by its corresponding glomerulus ($y_i$) and to be inhibited by the readout units ($x_k$). Eq (10b) requires the readout units to be directly excited by the mitral cells. This pattern of interaction between the mitral cells and the readout units implies an identification of the readout units $x_j$ with olfactory bulb granule cells, whose

main source of excitation is the mitral cells, which they in turn inhibit. Note that in our model the mitral cell/granule connections are symmetric. Since granule cells lack axons, the results of the computation must be read out from the mitral cell activations. This can be done by mirroring the integration of mitral cell input by granule cells as described in Methods Sec. 4.5, 'Cortical readout'.

## 2.2 Incorporating sister mitral cells

Eq (10a) indicates that each mitral cell $\lambda_i$ is inhibited by each granule cell $x_j$, of which there are hundreds of thousands in the mouse [20]. Thus, although the dynamics yield correct inference at convergence, if $A_{ij}$ is dense we are again faced with implausibly high connectivity. We take inspiration from the olfactory system to show how this problem can be addressed while still performing MAP inference.

So far we have assumed that each glomerulus provides input to one mitral cell (left panel of Fig 2), but in reality, each vertebrate mitral cell has several dozen 'sister' cells (mitral and tufted cells) that all receive input from the same glomerulus [21], (right panel of Fig 2), and all receive inhibitory feedback from granule cells. This suggests a way to reduce the number of mitral cell/granule cell connections: let each sister mitral cell connect to a different, non-overlapping, set of granule cells. Given that there are approximately 25–50 sister cells per glomerulus [22, 23], that would reduce connectivity by a factor of 25–50, yielding biologically plausible levels.

To derive circuitry that can implement this scheme, we start by assuming that the sister cells obey dynamics similar to Eq (10a),

$$\tau_\lambda \frac{d\lambda_i^s}{dt} = -\lambda_i^s + \frac{1}{\sigma^2}\left(y_i - S\sum_j W_{ij}^s x_j\right) \tag{11}$$

where $\lambda_i^s$ is the $s^{\text{th}}$ sister cell for glomerulus $i$ and $S$ is the number of sister cells. Below we will choose $W_{ij}^s$ to be zero for all but one $s$, which greatly reduces the number of granule cells that connects to each mitral cell, but for now we leave it arbitrary.

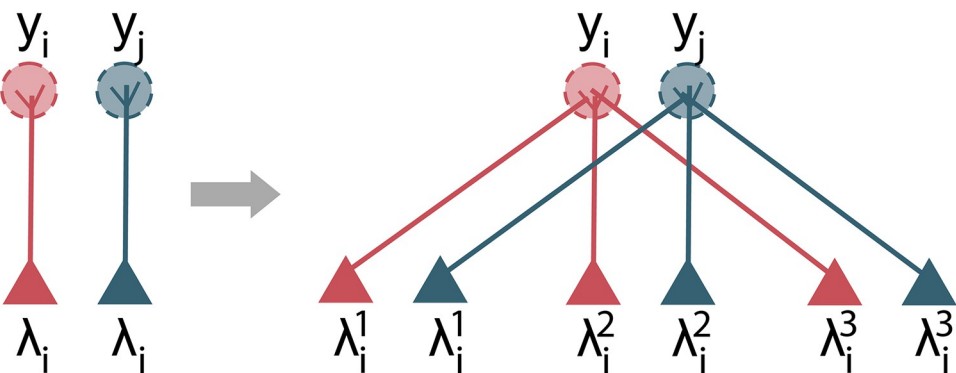

**Fig 2. Sister cells.** In the vertebrate olfactory bulb, each glomerulus, $y$, is sampled by not one (left panel) but approximately 25–50 'sister' mitral and tufted cells, $\lambda$ [22, 23] (right panel, with only three sisters to reduce clutter).

To see how sister mitral cells can perform correct MAP inference, note that the average sister cell activity,

$$\overline{\lambda}_i \equiv \frac{1}{S}\sum_s \lambda_i^s \tag{12}$$

evolves according to

$$\tau_\lambda \frac{d\overline{\lambda}_i}{dt} = -\overline{\lambda}_i + \frac{1}{\sigma^2}\left(y_i - \sum_j \sum_s W_{ij}^s x_j\right). \tag{13}$$

Letting the weights, $W_{ij}^s$, obey

$$A_{ij} = \sum_{s=1}^{S} W_{ij}^s, \tag{14}$$

the average sister cell activity evolves according to

$$\tau_\lambda \frac{d\overline{\lambda}_i}{dt} = -\overline{\lambda}_i + \frac{1}{\sigma^2}\left(y_i - \sum_j A_{ij} x_j\right). \tag{15}$$

This is identical to Eq (10a), the time evolution equation for $\lambda_i$, implying that if $\overline{\lambda}_i = \lambda_i$ at $t = 0$, then $\overline{\lambda}_i = \lambda_i$ for all time. Consequently, rather than computing $\lambda_i$ from Eq (10a), we can compute it by simply averaging over the sister mitral cells,

$$\lambda_i = \frac{1}{S}\sum_s \lambda_i^s. \tag{16}$$

We can, therefore, replace $\lambda_i$ in Eq (10b) with the right hand side of Eq (16), and, so long as the sister mitral cells evolve according to Eq (11), our model will implement MAP inference.

Eq (14) tells us only that the weights of the sister mitral cells add up to $A_{ij}$, but besides that we have complete freedom in choosing them. A trivial choice is $W_{ij}^s = A_{ij}/S$ (illustrated in Fig 3B). However, this doesn't help, as each sister mitral cell still receives $N$ inputs—one from each granule cell. What we want to do instead is make the connections sparse so that, as mentioned

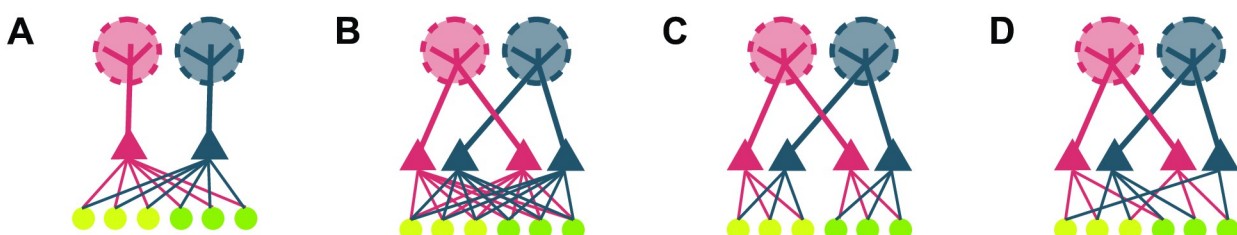

**Fig 3. Sparsifying connectivity with sister cells.** Each panel shows a schematic of the connectivity between mitral cells (triangles) and granule cells (circles). All connections are reciprocal. (A) Original scenario—no sister mitral cells. Connectivity is all-to-all, and the weights are $A_{ij}$. (B) A densely connected configuration where every granule cell connects to all sisters on each glomerulus, with weight $W_{ij}^s = A_{ij}/S$. The sister mitral cells receive the same number of connections as in panel A, but the granule cells receive twice as many as in panel A ($S$ times as as many in general). All-to-all connectivity is thus exacerbated. (C) Blocks of granule cells, indicated by the shades of green, connect to the same sister from each glomerulus, leading to maximally sparse connectivity. In this example, the first three granule cells connect to the first sister on each glomerulus, and the second three granule cells connect to the second sister; see Eq (17). Sister mitral cells now interact with a factor of 2 fewer granule cells ($S$ fewer in general), sparsifying mitral cell connectivity. (D) A more realistic maximally sparse connectivity pattern. Each granule cell connects to a single, randomly selected sister cell from each glomerulus; see Eq (18). Sister mitral cells connect to a factor of $S$ fewer granule cells on average, though individual mitral cells may connect to more (like the first green mitral cell) or fewer (like the second).

above, each sister cell receives input from a different, non-overlapping set of granule cells. If the sets are equal in size, this means each sister cell receives input from $N/S$ granule cells, an $S$-fold sparsification (Fig 3C and 3D).

There are many ways to make connectivity sparse. One is to divide granule cells into blocks, and then have the granule cells in each block project to one of the sister cells corresponding to each glomerulus,

$$s_j \sim U(\{1, \ldots, S\}) \tag{17a}$$

$$W_{ij}^s = \begin{cases} A_{ij} & \text{if } s = s_j, \\ 0 & \text{otherwise.} \end{cases} \tag{17b}$$

This scheme is shown in Fig 3C (with the blocks arranged sequentially). More realistic is for each granule cell to connect to a randomly selected sister cell from each glomerulus, so that $s_j$ also depends on glomerulus $i$,

$$s_{ij} \sim U(\{1, \ldots, S\}) \tag{18a}$$

$$W_{ij}^s = \begin{cases} A_{ij} & \text{if } s = s_{ij}, \\ 0 & \text{otherwise.} \end{cases} \tag{18b}$$

This scheme is shown in Fig 3D. In either case, each sister mitral cell now receives input from a factor of $S$ fewer granule cells. The granule cells still make $M$ connections (recall that $M$ is the number of glomeruli), but, at least in the olfactory system, $M$ is relatively small, on the order of $10^2$–$10^3$.

At this point we have demonstrated that the dynamics in Eqs (10b) and (11) will lead to MAP inference if $\lambda_i$ is set to the average of the sister mitral cell activity; that is, if Eq (16) is satisfied. However, no known cell performs the averaging required in Eq (16) and then projects to the granule cells, as required in Eq (10b). We therefore take an alternative strategy: we augment the dynamics to ensure that all the sister mitral cells converge to the average. To do that, we introduce a new cell type (which we will ultimately identify as periglomerular cells) that evolves according to

$$\tau_\mu \frac{d\mu_i^s}{dt} = \lambda_i^s - \overline{\lambda}_i. \tag{19}$$

This achieves the desired result: at equilibrium, when $d\mu_i^s/dt = 0$, all sister mitral cells associated with glomerulus $i$ have the same value—the average sister mitral cell activity. To ensure that $\mu_i^s$ converges to an equilibrium, rather than increasing or decreasing linearly with time, we need $\mu_i^s$ to couple to the sister mitral cells. A reasonable coupling is linear negative feedback, transforming Eq (10a) to

$$\tau_\lambda \frac{d\lambda_i^s}{dt} = -\lambda_i^s + \frac{1}{\sigma^2} \left( y_i - S \sum_{j=1}^{N} W_{ij}^s x_j - S\mu_i^s \right). \tag{20}$$

This certainly has the right flavor: positive $\mu_i^s$ tends to decrease $\lambda_i^s$, and vice versa, suggesting that all the sister mitral cells will eventually have the same value. But will they have the right value—the value implied by Eq (10a)? To answer that, we combine Eq (14) with Eq (20) to

write

$$\tau_\lambda \frac{d}{dt}\overline{\lambda}_i = -\overline{\lambda}_i + \frac{1}{\sigma^2}\left(y_i - \sum_{j=1}^{N} A_{ij}x_j - \sum_s \mu_i^s\right). \tag{21}$$

Except for the last term in parentheses, Eq (21) is exactly the same as the equation for $\lambda_i$, Eq (10a). Note, however, that

$$\tau_\mu \frac{d}{dt}\sum_s \mu_i^s = S(\overline{\lambda}_i - \overline{\lambda}_i) = 0. \tag{22}$$

Hence, if we initialize $\mu_i^s$ so that $\sum_s \mu_i^s$ is zero, it will remain zero for all time. In that case, the equation for the sister cell average, Eq (21), is identical to Eq (10a). Consequently, each of the sister cells converge to the correct mean, and so we can replace Eq (10b) with

$$\tau_x \frac{dx_j}{dt} = -\frac{\partial\phi}{\partial x_j} + \sum_{i,s} W_{ij}^s \lambda_i^s. \tag{23}$$

Thus, under the dynamics given in Eqs (19), (20) and (23), with $W_{ij}^s$ obeying Eq (14), the network performs MAP inference.

The variables $\mu_i^s$ in Eq (19) are driven by a weighted average of sister cell activations. The observed backpropagation of mitral cell action potentials to the glomeruli [24, 25] and the electrical coupling of sisters at the glomeruli [26] might contribute to the neural implementation of just such an average. Thus we have provisionally identified the $\mu_i^s$ variables with olfactory bulb periglomerular cells because they inhibit the mitral cells and are in turn excited by them [27, 28], and do not receive direct receptor input themselves. Periglomerular interneurons constitute a diverse group of cells [28, 29] and there is currently limited insight into their detailed wiring diagram [28, 30]. Nevertheless, the type of cell described above (reciprocal connections with mitral/tufted cells without direct receptor input) is reminiscent of the Type II periglomerular cells of Kosaka and Kosaka [29, 31] (see also [32]).

To summarize, the introduction of sister cells allows exact MAP inference to be performed while reducing, by a factor of $S$, the number of granule cells each mitral cell must connect to. It is in this sense that sister cells allow MAP inference to be performed with sparse connectivity.

## 2.3 Leaky periglomerular cells

The dynamics in Eq (19) implies that the periglomerular cells, $\mu_i^s$, do not leak; i.e., they are perfect integrators. This is at odds with biology, since we imagine that integration is performed by neuronal membranes, and neuronal membranes are leaky [33]—though periglomerular cells may be less leaky than most other neurons [28, 34]. We can introduce a leak term into the dynamics,

Periglomerular cell activity relative to baseline, with leak :
$$\tau_\mu \frac{d\mu_i^s}{dt} = -\varepsilon\mu_i^s + \lambda_i^s - \overline{\lambda}_i \tag{24}$$

where $\varepsilon$ sets the magnitude of the leak. One advantage of introducing this leak is that we no longer have to worry about initializing the $\mu_i^s$ so that their mean is zero, since with a leak term the mean periglomerular activation decays to zero,

$$\tau_\mu \frac{d\overline{\mu}_i}{dt} = -\varepsilon\overline{\mu}_i + \overline{\lambda}_i - \overline{\lambda}_i = -\varepsilon\overline{\mu}_i. \tag{25}$$

The price we pay is that the system no longer computes the MAP solution exactly. As we show in

Methods, Sec. 4.1, when there is leak the system of equations minimizes the wrong objective,

$$\mathcal{L}_\varepsilon(\mathbf{x}) = q_\varepsilon \left[ \sum_{j=1}^N \phi(x_j) + \frac{1}{2\sigma^2} \sum_{i=1}^M \left( y_i - \sum_j A_{ij} x_j \right)^2 \right]$$
$$+ (1 - q_\varepsilon) \left[ \sum_{j=1}^N \phi(x_j) + \frac{1}{2\sigma^2} \sum_{i=1}^M \sum_{s=1}^S \frac{1}{S} \left( y_i - S \sum_j W_{ij}^s x_j \right)^2 \right]$$

(26)

where

$$q_\varepsilon \equiv \frac{S}{S + \varepsilon \sigma^2} .$$

(27)

In the limit of no leak ($\varepsilon \to 0$, so that $q_\varepsilon \to 1$), we recover the correct objective (compare to Eq (7)). For non-zero leak, the objective differs from the MAP objective, so solutions will differ from the MAP solution. However, as we show numerically in Sec. 2.5.2, for biologically relevant values of $\varepsilon$ these deviations are small.

Note that as the number of sister mitral cells $S$ increases, $q_\varepsilon$ approaches 1. Naively, this suggests that we should recover the true objective in the large $S$ limit. However, this naive expectation ignores the fact that there is a factor of $S$ in the second term in Eq (26); in the large $S$ limit, this cancels the $1/S$ dependence in $(1 - q_\varepsilon)$. Consequently, it is not immediately clear how inference depends on $S$ when there is nonzero leak. We addressed this numerically, and found that the error relative to the MAP solution increases monotonically with the number of sisters; see Sec. 2.5.2.

## 2.4 Implementation in neural circuitry

The mitral and periglomerular cell dynamics (Eqs (20) and (19), respectively) are in a form suitable for implementation by neural circuitry. However, the granule cell dynamics in Eq (23) cannot be implemented directly because of the presence of not-everywhere-differentiable terms in the prior, $\phi$ (see Eq (6)). We thus implement related, neurally plausible, dynamics that has the same fixed point. Specifically, we note that at the fixed point of Eq (23), $x_j$ satisfies

$$\frac{1}{\beta} \left( \sum_{i,s} W_{ij}^s \lambda_i^s - \gamma x_j \right) \in \partial(|x_j| + \mathbb{I}(x_j \geq 0))$$

(28)

where $\partial$ is the subgradient operator [35]. If at the solution $x_j > 0$, then the subgradient operator reduces to the ordinary gradient and yields the value 1, and we have

$$x_j = \frac{1}{\gamma} \left( \sum_{i,s} W_{ij}^s \lambda_i^s - \beta \right).$$

(29)

On the other hand, when $x_j = 0$ the subgradient is the set $(-\infty, 1]$, so we have

$$\frac{1}{\beta} \sum_{i,s} W_{ij}^s \lambda_i^s \in (-\infty, 1] ,$$

(30)

which we can write as an inequality,

$$\sum_{i,s} W_{i,j}^s \lambda_i^s \leq \beta.$$

(31)

Thus, $x_j = 0$ whenever Eq (30) is satisfied; combining that with Eq (29) (which is valid when $x_j > 0$), we have

$$x_j = \frac{1}{\gamma}\left[\sum_{i,s} W_{ij}^s \lambda_i^s - \beta\right]^+ \tag{32}$$

where $[\cdot]^+$ is the threshold-linear operator. A neural implementation of this function would have $x_j$ responding instantaneously to changes in the mitral cell activations $\lambda_i^s$, which is implausible. Instead we employ a membrane voltage variable $v_j$ which integrates the mitral cell input and interpret $x_j$ as the resulting firing rate. The full set of equations describing the model is, therefore,

$$\text{MC activity}: \qquad \tau_\lambda \frac{d\lambda_i^s}{dt} = -\lambda_i^s + \frac{1}{\sigma^2}\left(y_i - S\sum_j W_{ij}^s x_j - S\mu_i^s\right) \tag{33a}$$

$$\text{PGC activity}: \qquad \tau_\mu \frac{d\mu_i^s}{dt} = -\varepsilon\mu_i^s + \lambda_i^s - \overline{\lambda}_i \tag{33b}$$

$$\text{GC voltage}: \qquad \tau_v \frac{dv_j}{dt} = -v_j + \sum_{i,s} W_{ij}^s \lambda_i^s \tag{33c}$$

$$\text{GC firing rate}: \qquad x_j = \frac{1}{\gamma}[v_j - \beta]^+ \tag{33d}$$

where the weights $W_{ij}^s$ satisfy Eq (14). Eqs (33a) and (33b) correspond to Eqs (20) and (19), respectively, and Eqs (33c) and (33d) implement Eq (32). The sparsity parameters $\beta$ and $\gamma$ from the prior, Eq (6) appear as the threshold and inverse gain of the granule cell firing rate, Eq (33d). Because these parameters reflect the statistics of the environment, we assume that they can be set appropriately on evolutionary time scales as the species adapts to its environment. But they can also be adjusted on faster time scales by, for example, using cortical feedback to add a background voltage to the bracketed term in Eq (33d), modifying the threshold of the granule cell firing rate, and so altering the sparsity prior. We leave the investigation of such possibilities to future work.

A circuit that implements these equations is shown schematically in Fig 4. As promised, each mitral cell interacts with only a subset of the granule cells, as in Fig 3D. This reduces mitral-granule connectivity by a factor of $S$ (though the *total* number of mitral-granule synapses stays the same due to the introduction $S$ sister mitral cells per glomerulus). The information from the other granule cells is delivered indirectly to each mitral cell via the influences of the glomerular average of the sister cell activations and periglomerular inhibition.

## 2.5 Simulations and linear analysis

To investigate the behaviour of the system, we performed a series of simulations using the model summarized in Eq (33). The three questions that guided our choice of simulations, along with brief answers, are:

1. What do the dynamics of sister cells look like? Our analysis so far shows only that if the dynamics converges, it yields the MAP solution. However, we have not shown that the dynamics necessarily converges, or said anything about transient behaviour. Thus the first goal of our simulations is to check convergence empirically, and to qualitatively assess the transient dynamics and its biological plausibility.

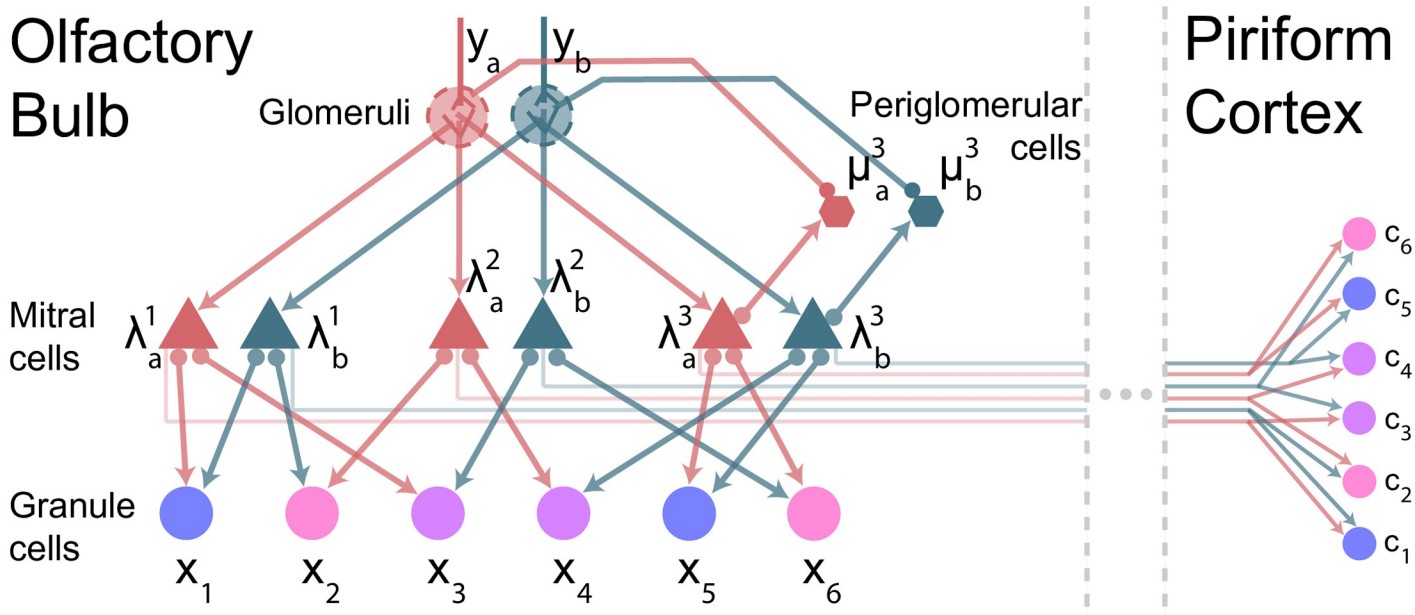

**Fig 4. Sparsely connected inference circuit, with readout in piriform cortex.** Arrows and filled circles indicate excitatory and inhibitory connections, respectively. Each sister cell $\lambda_a^s$ receives input $y_a$ from a glomerulus and interacts with a subset of the granule cells $x_j$, reducing connectivity by a factor of $S$ ($S = 3$ in this example). Information is shared between granule cells through the periglomerular cells $\mu_a^s$ (only ones with $S = 3$ are shown, for clarity). Those cells receive excitatory input from the mitral cells and inhibitory input corresponding to the average sister cell activity (as required by Eq (33b)). The inhibitory input comes from back-propagating action potentials, which travel along mitral cell apical dendrites and are available at the glomeruli [24, 25]. The result of the inference is simultaneously available in granule cells and neurons in piriform cortex (see Methods, Sec. 4.5, 'Cortical readout').

Answer: we show numerically that the dynamics does indeed converge, and show analytically that solutions are stable for the parameter regime of interest.

2. What is the effect of non-zero periglomerular leak ($\varepsilon > 0$)? The dynamics in Eq (33) yields the MAP solution at convergence only when the periglomerular cells have zero leak, yet any biological implementation of these dynamics will have non-zero leak. It is important to determine the extent to which realistic values of the leak affect the dynamics and the inference solutions.
Answer: for realistic values of the leak, the effect on MAP inference is small.

3. Finally, how do the various parameters affect the transient dynamics, and which parameters are most important? In particular, does the dynamics become qualitatively unrealistic when some parameters are changed within biologically reasonable ranges?
Answer: the transient dynamics is extremely robust to parameters, and exhibits very little change over a broad range.

Below we expand on these answers.

**2.5.1 System dynamics with sister cells.** In our simulations we used the base parameters given in Table 1; departures from those parameters will be flagged. Sister cell connectivity was set according to Eq (18); see Methods Sec. 4.4 for further details.

Fig 5 shows typical activity patterns for mitral, periglomerular, and granule cells. Panel A shows the response of all four sister mitral cells from a representative glomerulus, and panel B shows the response of the corresponding periglomerular cells. Although there is some initial variability in the responses—in particular decaying oscillations—the sister cells converge to the same value, as expected. Panel C shows granule cell activity, and demonstrates that the

**Table 1. Base parameters used in the simulations.**

| Parameter | Description | Base Value |
|:---:|:---|:---:|
| $M$ | Number of glomeruli | 50 |
| $N$ | Number of granule cells | 1200 |
| $S$ | Number of sister cells per glomerulus | 4 |
| $n$ | Number of components present in the true odour | 3 |
| $\sigma^2$ | Receptor variance | $10^{-2}$ |
| $\beta$ | $\ell_1$ penalty | 3 |
| $\gamma$ | $\ell_2$ penalty | 1 |
| $\tau_\nu$ | Granule cell membrane time constant | 35 ms |
| $\tau_\lambda$ | Mitral cell membrane time constant | 50 ms |
| $\tau_\mu$ | Periglomerular cell membrane time constant | 35 ms |
| $\varepsilon$ | Periglomerular cell leak | 0 |
| $A_{ij}$ | Elements of the affinity matrix | $\mathcal{N}(0, 1/M)$ |

readout converges to the MAP solution within a few hundred milliseconds. This is reflected in the root mean square (RMS) error between the granule cell responses and the MAP estimate, which decays exponentially (panel D).

In Fig 6 we show typical dynamics when there are $S$ = 1, 8 and 25 mitral cells per glomerulus. In all cases, the granule cells converge to the MAP solution within a few hundred milliseconds. The main difference between the three values of $S$ is that when $S$ = 1, convergence to the MAP solution is slightly faster than when $S > 1$, as indicated by the slightly steeper RMS error curves in the bottom left panel. Otherwise, the dynamics in all three cases is qualitatively similar.

**Linear analysis**. Given that our network was designed to perform MAP inference, the asymptotic behavior shown in Figs 5 and 6 is relatively unsurprising. However, our analysis so far says nothing about the transient behavior, which, as can be seen in Figs 5 and 6, is characterized by large oscillations. To understand this behavior—in particular how stability and oscillation frequency depend on the parameters, including the number of sister mitral cells—we need to analyze the dynamics.

Because of the rectifying nonlinearity, that is hard to do exactly. However, our simulations so far (in particular the granule cell activity in Figs 5 and 6) suggest that the composition of the 'active' set of granule cells stabilizes before the dynamics terminates. Once this active set has stabilized, so that the rectifications remain within the linear regime, the granule cell activations $x_j$ can be replaced by their corresponding voltage variables $v_j$, and the system becomes linear. We thus performed linear analysis of Eq (33) around a solution with $n$ active granule cells, and used the results to both explain the transient behavior we have seen so far, and guide further investigation of the system.

The linearized dynamics relative to their input-dependent fixed-point for $n$ active granule cells is given by (Methods, Sec. 4.2)

$$\tau_\lambda \frac{d\delta\lambda_i^s}{dt} = -\delta\lambda_i^s - \frac{S}{\gamma\sigma^2}\sum_{j=1}^{n} W_{ij}^s \delta v_j - \frac{S}{\sigma^2}\delta\mu_i^s \tag{34a}$$

$$\tau_\mu \frac{d\delta\mu_i^s}{dt} = -\varepsilon\delta\mu_i^s + \delta\lambda_i^s - \frac{1}{S}\sum_{s=1}^{S}\delta\lambda_i^s \tag{34b}$$

$$\tau_\nu \frac{d\delta v_j}{dt} = -\delta v_j + \sum_{i=1}^{M}\sum_{s=1}^{S} W_{ij}^s \delta\lambda_i^s \tag{34c}$$

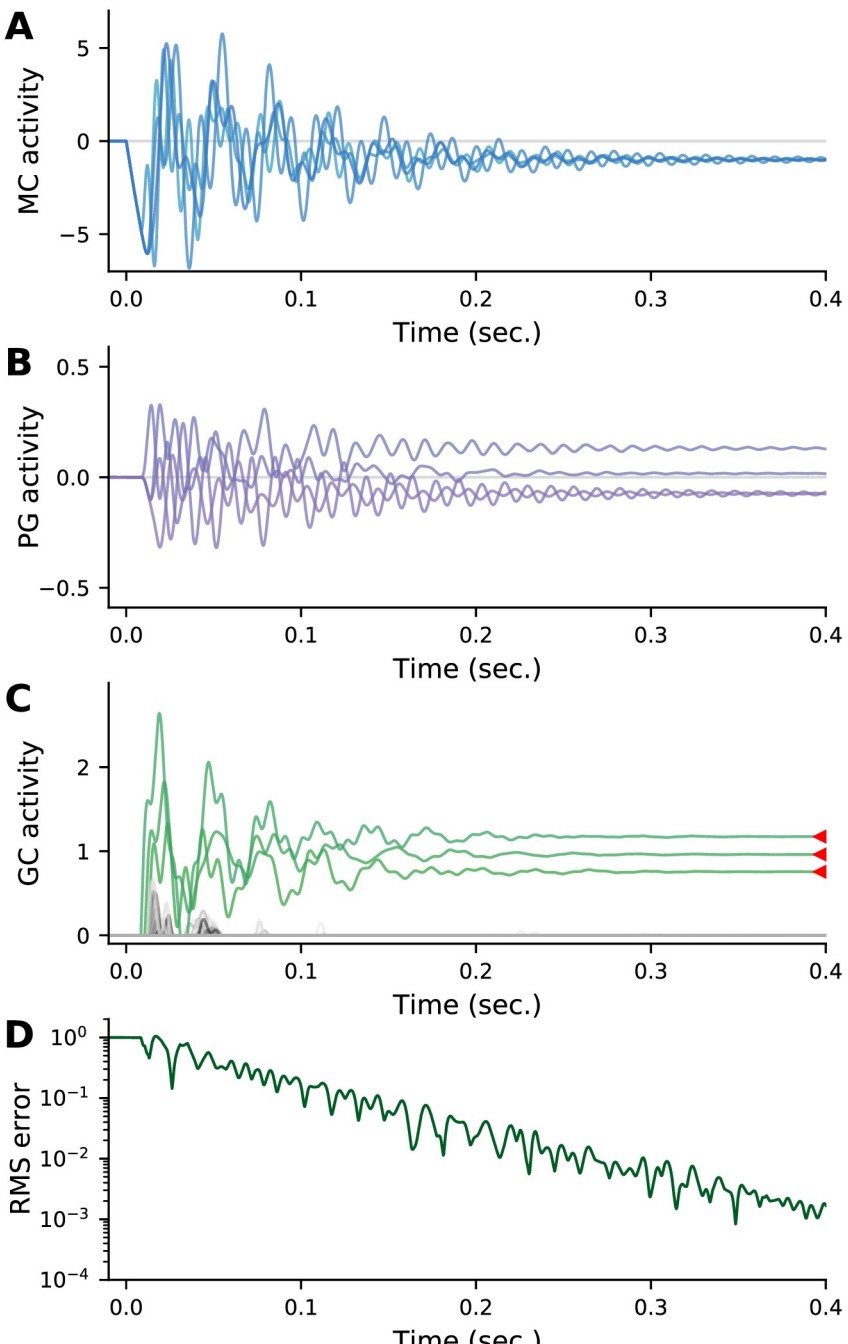

**Fig 5. Example dynamics for a network with base parameters (Table 1).** (A) All four sister mitral cells from a representative glomerulus. Although the sister cells start off with identical activations, their activities quickly diverge due to their differing connectivities to the granule cells. Ultimately, however, they converge to the same value. (B) Periglomerular cells from the glomerulus in panel A. (C) Granule cells. Red arrows indicate the MAP solution for the three components present in the odour. Granule cells representing these components are shown in green, the others in gray. After an initial period of activity, the system settles into the MAP solution. (D) Time course of the root-mean-square error between the granule cell activations and the MAP solution normalized to its initial value, indicating convergence to the MAP solution.

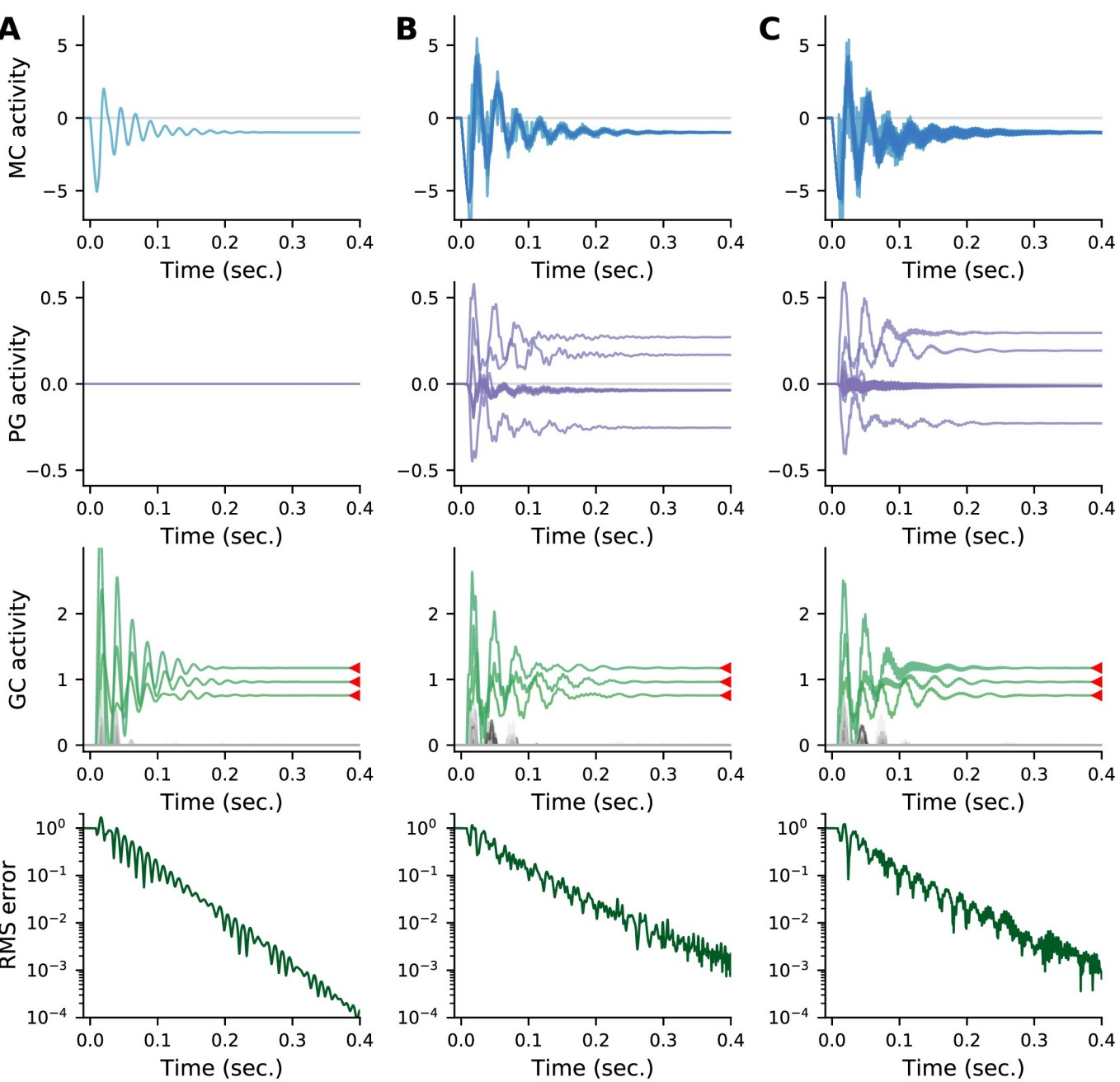

**Fig 6. Typical dynamics as the number of mitral cells per glomerulus, $S$, is varied relative to the base model given in Table 1.** (A) $S = 1$, (B) $S = 8$, and (C) $S = 25$ mitral cells per glomerulus, compared to $S = 4$ in Fig 5. Note the lack of periglomerular cell activity and slightly faster convergence for $S = 1$. Otherwise, the dynamics is qualitatively similar for all values of $S$, including convergence to the MAP solution.

where the $\delta$ in front of each variable indicates a small deviation from the fixed point. Note that we replaced the granule cell activations $x_j$ by their membrane voltages $v_j$, as motivated above, and that the indexing of the latter variables is over the active set of $n$ granule cells.

To solve these equations, we let the dynamical variables have the time dependence $e^{\xi t}$, which results in an eigenvalue equation for $\xi$. That equation can't be solved exactly (at least not

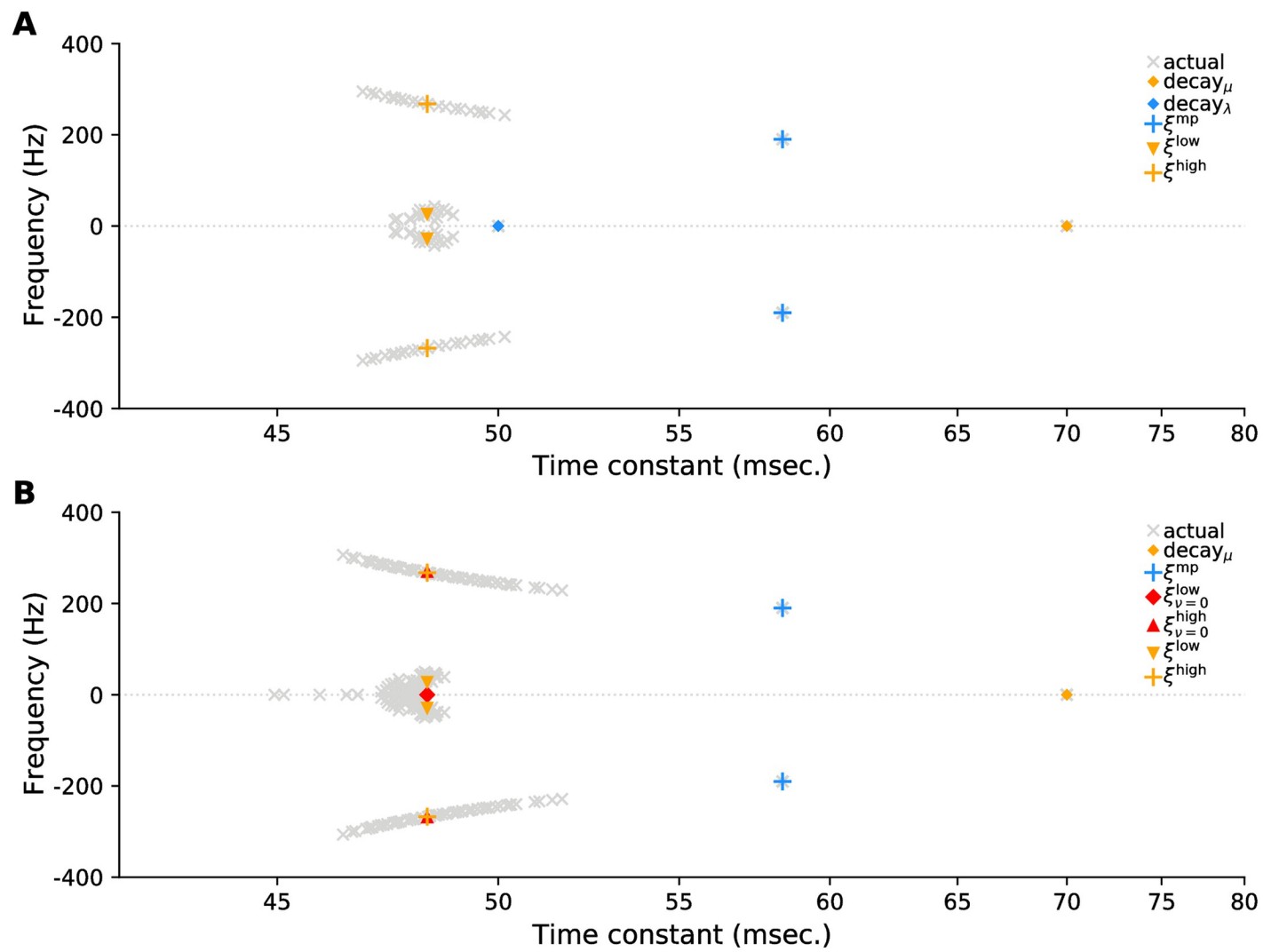

**Fig 7. Eigenvalues of the linearized system computed numerically (gray crosses) and analytically (coloured symbols).** Eigenvalues labeled 'decay$_\mu$', Eq (57), 'decay$_\lambda$', Eq (58) and '$\xi^{mp}$', Eq (59), are exact; those with the superscripts 'low' or 'high', Eq (76), are approximate. (A) $n = 20$ active granule cells. (B) $n = 55$, so that $n >$ $M$. In both panels, blue eigenvalues are for modes involving only mitral and periglomerular cells, modes for orange and red eigenvalues also involve granule cells. The time constant ($x$-axis) and frequency ($y$-axis) for each eigenvalue $\xi$ are $-1/\text{Re}(\xi)$ and $\text{Im}(\xi)/2\pi$, respectively. We used large $n$ to better show the spread of eigenvalues. Parameters from Table 1, except $S = 25$ and $\varepsilon = 1/2$, and $n$ depends on the panel. See Sec. 4.2 for additional details.

for all eigenvalues), but the approximate eigenvalues, derived in Methods, Sec. 4.2, are reasonably close to the true ones, as shown in Fig 7. In that figure, eigenvalues near blue markers are for modes involving only mitral and periglomerular cells ($\delta v_j = 0$), while eigenvalues near orange and red markers are for modes involving granule cells as well ($\delta v_j \neq 0$).

Each gray cross in Fig 7 corresponds to a mode of the system, and near the equilibrium the activity consists of a sum of these modes. However, because the modes cluster, the system admits only a handful of behaviors, which we summarize as follows:

1. Low frequency oscillations, '$\xi^{low}$', whose frequency does not change with added sisters (see Methods, Eq (80));

2. Two high frequency oscillations, '$\xi^{\text{high}}$' (Methods, Eq (78)), which involves all cell types, and '$\xi^{\text{mp}}$' (Methods, Eq (59)), which involves only the mitral and periglomerular cells. The frequency of these oscillations increases with added sisters;

3. Purely decaying modes (no oscillations), 'decay$_\mu$', (orange diamond in Fig 7; see Methods, Eq (57)), which has a decay rate that is proportional to $\varepsilon$, and 'decay$_\lambda$' (Methods, Eq (58)), which is present only when $n < M$ (blue diamond in the top panel of Fig 7). The latter involves the mitral and periglomerular cells, but not the granule cells.

Fig 7 shows the eigenvalue spectrum for only one set of parameters. What about other choices? The time constants, $\tau_\lambda$, $\tau_\mu/\varepsilon$ and $\tau_\nu$, set the time scale for decay. The other relevant parameters are the number of sister mitral cells, $S$, and $\gamma$ and $\sigma^2$; the latter two appearing in Eq (34a). As we show in Methods, Sec. 4.2, these have two main effects. First, the oscillation frequencies $\xi^{\text{high}}$ and $\xi^{\text{mp}}$ scale as $\sqrt{S/\sigma^2\gamma}$ (Eq (78)) and $\sqrt{S/\sigma^2}$ Eq (59), respectively, as shown in Fig 8A and 8B. The second effect is on the decay time constants, which are mainly independent of $S$, except when $S = 1$; in that case there is no $\xi^{\text{mp}}$ mode, which can affect the decay rates (see Fig 8C). For a detailed analysis of the effect of parameters on the transient behavior, see Methods Sec. 4.2.

The linear analysis can also tell us whether the MAP equilibrium can be unstable. We show in Methods, Sec. 4.2.2 that it is stable in the parameter regime of interest, which is $\sigma^2 \ll 1$; we did not investigate stability when $\sigma^2$ isn't especially small.

**2.5.2 The effect of periglomerular leak.** As discussed in Sec. 2.3, our network performs MAP inference only if the periglomerular leak, $\varepsilon$, is zero. What happens in the realistic case, when it's not zero? Here we address that question through simulations. From Eq (33b), we see that the effective time constant of the periglomerular cells is $\tau_\mu/\varepsilon = 35$ ms/$\varepsilon$ (see Table 1), so relevant values of $\varepsilon$ are near 1.

In Fig 9 we show typical dynamics for $\varepsilon = 1$ and $\varepsilon = 2$. Non-zero values of the leak mean the system no longer performs MAP inference; that's reflected in the plateauing of RMS error

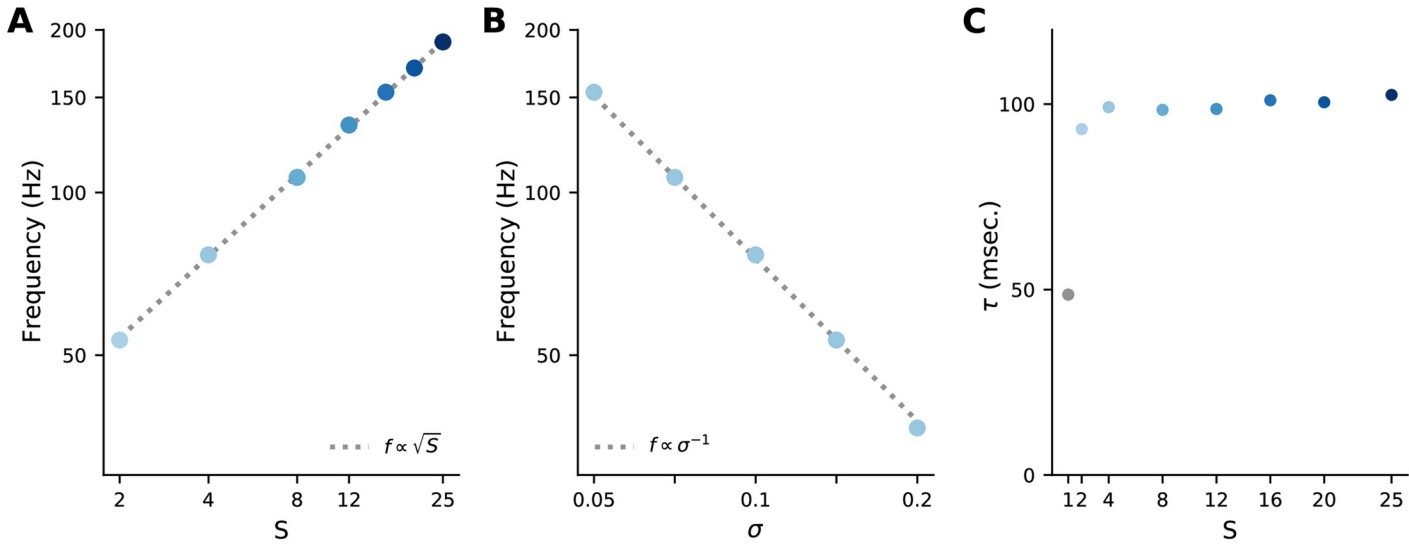

**Fig 8. Transient properties of inference dynamics.** (A) Dependence of the high-frequency mode of the mitral cell responses on the number of sisters, $S$. (B) Dependence on the standard deviation of the noise, $\sigma$. In both panels, the dashed lines are fits predicted by the linear analysis. (C) Dependence of the mitral cell decay time constant on the number of sisters $S$. Going from $S = 1$ to $S = 2$ sisters per glomerulus decreases the decay time constant by a factor of about 2. A large change in decay rate from $S = 1$ to 2, followed by a slower change for $S \geq 2$, is typical, although the details are parameter-dependent. Other parameters as in Table 1.

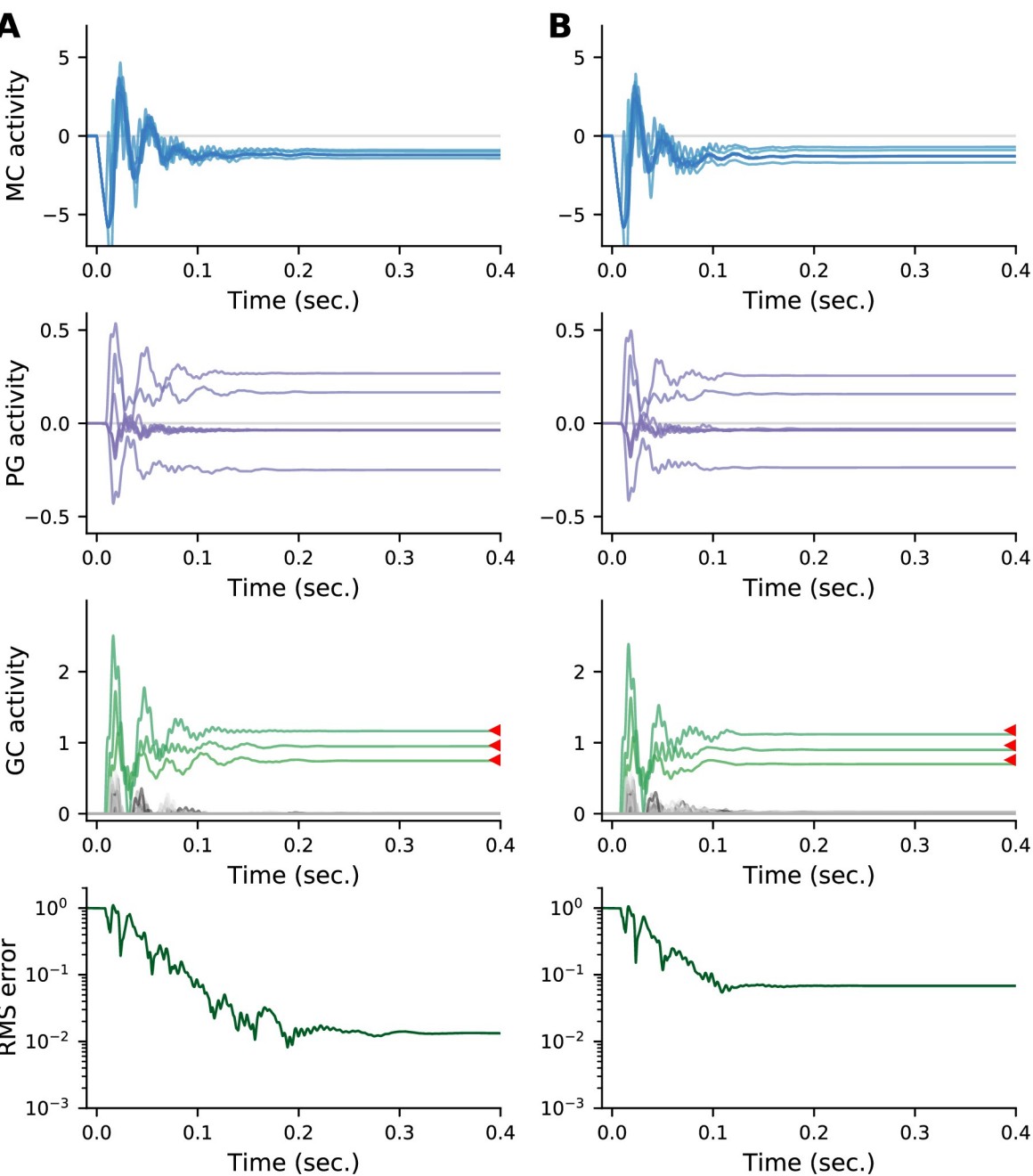

**Fig 9. Effect of periglomerular leak $\varepsilon$ on the dynamics.** Parameters from Table 1, but with the number of sisters $S$ fixed at 8 and the leak at (A) $\varepsilon = 1$ and (B) $\varepsilon = 2$. Note that sister mitral cells no longer converge to the same value (top row). Increasing the leak results in higher RMS error relative to the MAP estimate (red triangles).

relative to the MAP solution. In both cases however, the effect on granule cell activity is small. Non-zero leak also means that the periglomerular cells are no longer able to force sister cells to the same value at convergence. This is increasingly visible as the leak increases (compare granule cell activity at convergence in panels A and B of Fig 9). Again, though, the effect is small.

As discussed in Sec. 2.3, the effect of the periglomerular leak depends on the number of sister mitral cells, $S$. In Fig 10A we plot the asymptotic RMS error versus $\varepsilon$ as we vary $S$.

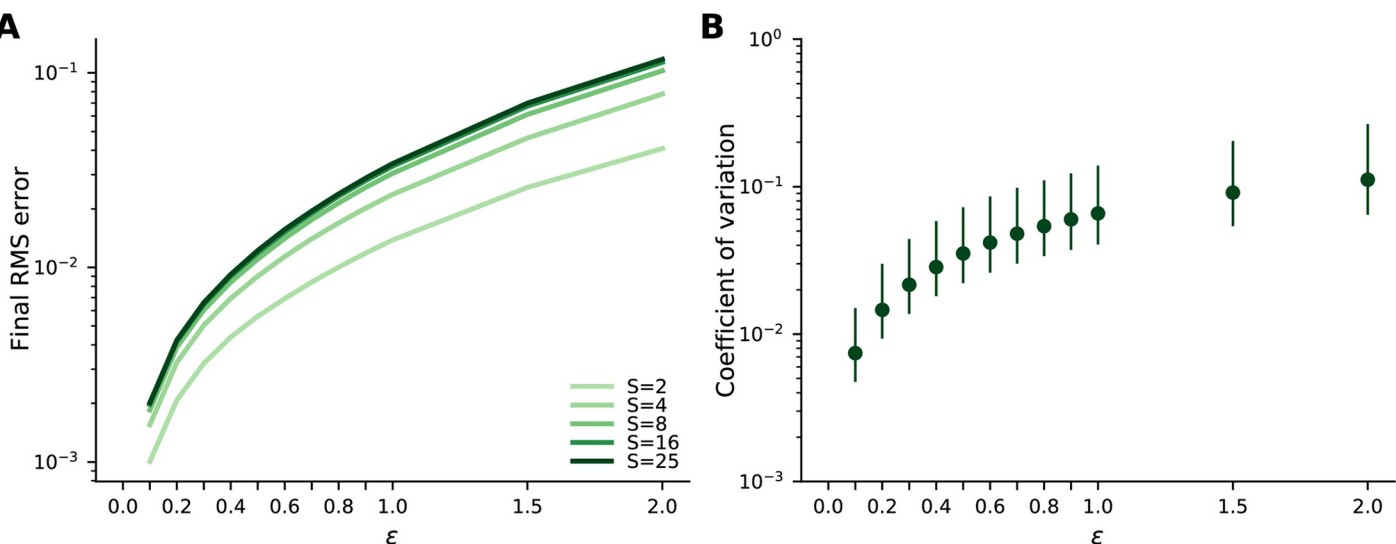

**Fig 10. Properties at convergence versus periglomerular leak $\varepsilon$.** (A) Mean RMS error at convergence, relative to its initial value, as a function of leak for different numbers of sister cells per glomerulus. The RMS error increases with leak and with the number of sister cells, but never gets very large. Means are computed over 10 different odours and 5 different olfactory bulbs. (B) Median (dots) and inter-quartile range (lines) of the coefficient of variation (CV; standard deviation divided by the mean) of the ratios of sister cell activity at convergence as a function of the leak parameter $\varepsilon$. CVs are computed over the ratios of all unique pairs of 25 sisters in a glomerulus, percentiles are computed over all glomeruli in response to 10 different odours across 5 different olfactory bulbs. Median CVs remain low even for high values of leak. Parameters from Table 1, except for $S$ and $\varepsilon$.

Increasing the number of sisters increases the steady state RMS error relative to the MAP solution, but past about $S = 8$ the number of sisters has very little effect on the error. In Fig 10B we plot the spread in sister cell activity relative to the mean for $S = 25$, showing that it remains small for large values of the leak.

**2.5.3 Robustness.** Finally, we examined how the parameters affect the dynamics in the non-leaky setting. Because the system always arrives at the MAP solution, we focus on the transient dynamics, and in particular on whether the system remains within a biologically plausible range.

**Number of odour components, $n$.** Intuitively, we expect that as the number of odour components, $n$, increases, the inference problem will become harder. In Fig 11A we show an example with 10 odour components present, and indeed we see that inference (blue lines) is not great: although the odour components that are present are correctly inferred, many odour components that are not present are inferred as well. Fig 11B corroborates this: the number of recovered odour components exceeds the number of true ones, $n$, when $n$ is sufficiently large. Note, though, that as $M$ increases, the system can accurately infer more odours. Moreover, as can be seen in Fig 11C, for all values of $n$ tested the dynamics converges to the MAP solution at similar rates.

**Number of cells.** So far our simulations have used $M = 50$ glomeruli and $N = 1200$ granule cells. However, most olfactory systems are much larger than this (we used smaller populations solely to speed up simulations). For example, in the fly $M \approx 50$ and $N \approx 2500$, while in the mouse, $M \approx 1000$ and $N \approx 10^6$. In Fig 12 we show circuit dynamics for larger systems; up to $M = 200$ and $N = 4800$. Consistent with the fact that the linear analysis (Methods, Sec. 4.2) doesn't predict a strong dependence on size, the dynamics are qualitatively similar to our simulations with small $M$ and $N$, and the MAP solution is achieved within a similar time frame.

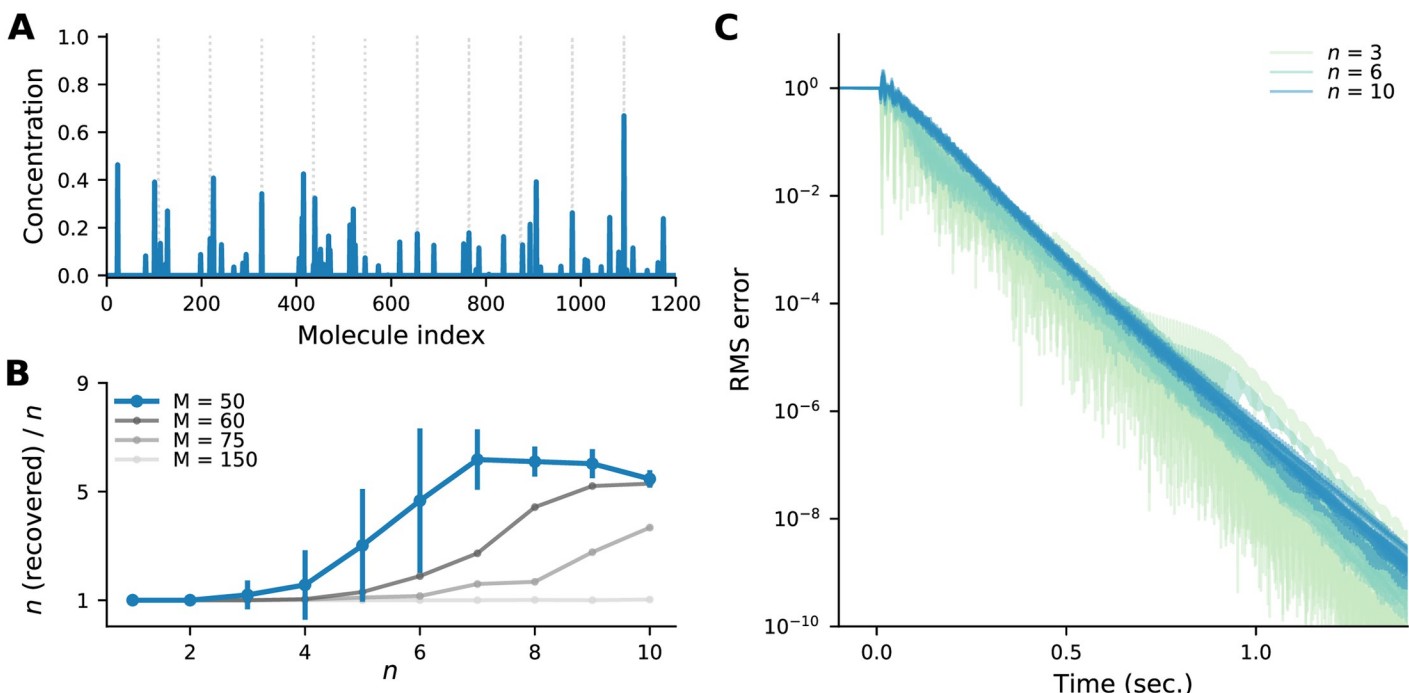

**Fig 11. The effect of the number of odour components, *n*, on inference.** (A) The MAP estimate for an odour with $n = 10$ components contains many more than 10 non-zero values. (B) Ratio of the number of odour components inferred using MAP inference to the true number, versus the true number of odour components, $n$ for different numbers of glomeruli, $M$. An odour component was considered inferred when granule cell activation was greater than $10^{-2}$. Dots are averages over 12 trials, vertical lines are standard deviations (shown only for $M = 50$ for clarity). When using the default number of glomeruli, $M = 50$, extra odour components are inferred when $n$ is above about 3. Adding glomeruli, however, allows more odours to be correctly inferred. (C) Time course of the RMS error between the granule cell activations and the MAP estimate, for 6 different trials at each $n$. Parameters from Table 1, except $S = 8$, and $M = 50$ unless indicated otherwise.

## 2.6 Relative sister cell activity at convergence

Our proposed scheme makes a strong experimental prediction about the relative activity of sister mitral cells at convergence, but it takes some analysis to determine exactly what that prediction is. According to Eq (33b), when there is no periglomerular leak ($\varepsilon = 0$), all sister mitral cells converge to the same value—the mean, $\overline{\lambda}_i$, which depends only on which odours were presented. However, exact equality relies on a particular choice of coordinates. Suppose, for instance, that the experimentally recorded mitral cell firing rates, denoted $\tilde{\lambda}_i^s$, were related to those in our model, $\lambda_i^s$, by cell-specific invertible transformations $f_i^s$,

$$\lambda_i^s = f_i^s(\tilde{\lambda}_i^s). \tag{35}$$

(Invertibility is required because there must be a one-one mapping between trajectories in the transformed and non-transformed variables). Because our model predicts that at convergence $\lambda_i^s = \overline{\lambda}_i$, independent of odour, it follows that at convergence,

$$f_i^s(\tilde{\lambda}_i^s) = f_i^{s'}(\tilde{\lambda}_i^{s'}) . \tag{36}$$

This prediction would be hard to verify experimentally, because it requires knowledge of the transformations $f_i^s$ and $f_i^{s'}$. However, if we assume that the transformations are differentiable, we can arrive at a more useful expression by differentiating both sides with respect to $\tilde{\lambda}_i^s$ and

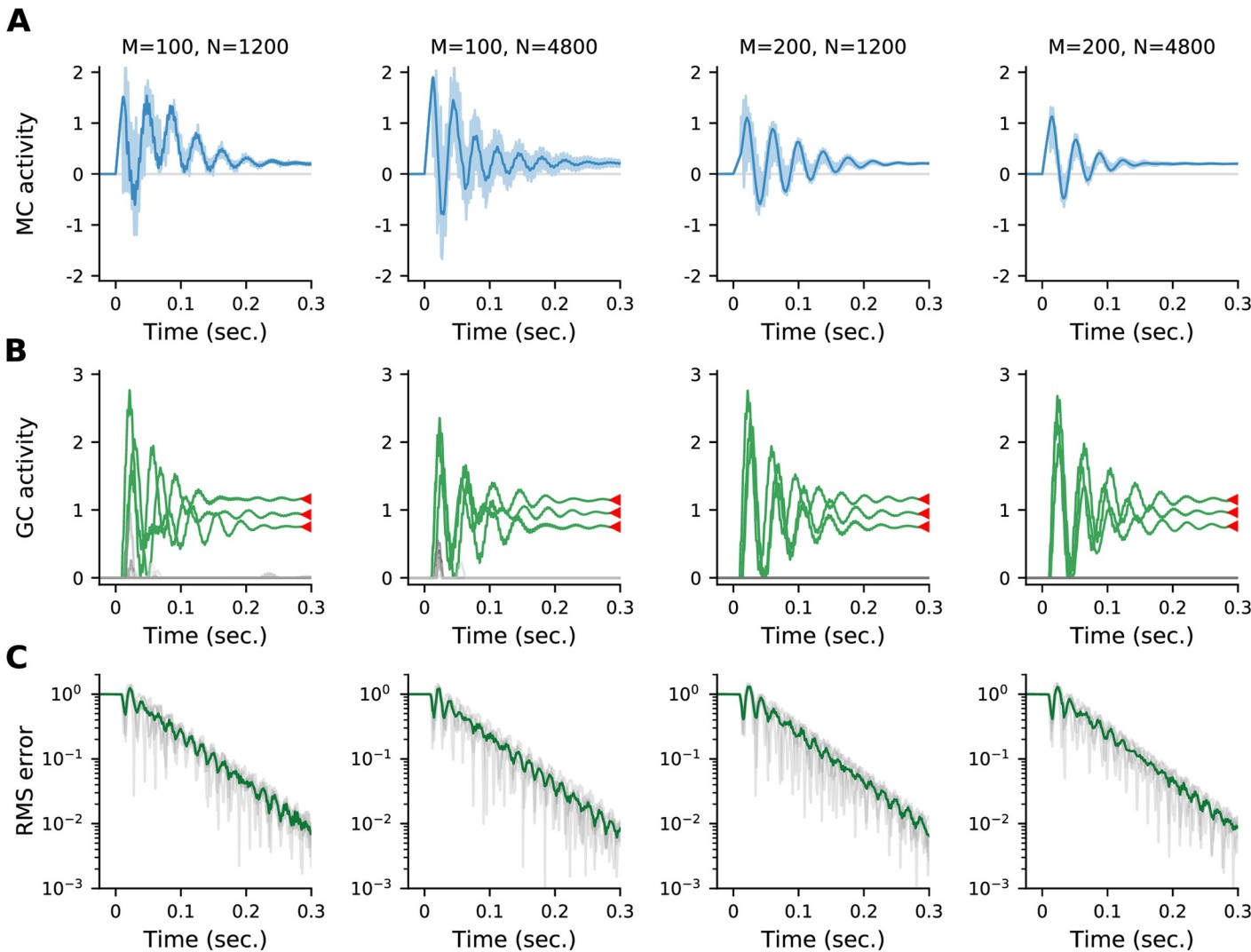

**Fig 12. Dynamics for different numbers of glomeruli, *M*, and granule cells, *N* (specified at the top of each column), are qualitatively similar.** (A) Mitral cells. The activity of one mitral cell is highlighted with a dark trace; the remaining sisters are overlayed with a light trace. (B) Granule cells. Granule cells representing molecules present in the true odour are coloured green and the 10 most active granule cells representing molecules *not* present in the true odour are coloured gray. Granule cell dynamics are all qualitatively similar and converge to the MAP solution (red triangles). (C) RMS error between the granule cell activity and the MAP estimate for 5 random systems of the same size as in the previous rows (gray) and the average (green). All systems converge with similar dynamics to the MAP estimate. Remaining parameters are from Table 1, except *S* = 25.

rearranging terms,

$$\frac{d\tilde{\lambda}_i^s}{d\tilde{\lambda}_i^{s'}} = \frac{\partial f_i^{s'}/\partial \tilde{\lambda}_i^{s'}}{\partial f_i^s/\partial \tilde{\lambda}_i^s} \; . \tag{37}$$

Because $f_i^s(\tilde{\lambda}_i^s)$ and $f_i^{s'}(\tilde{\lambda}_i^{s'})$ are invertible and differentiable, and thus monotonic, functions of their arguments, the sign of the derivatives are independent of the value of either $\tilde{\lambda}_i^s$ or $\tilde{\lambda}_i^{s'}$. This means that the sign of the right hand side is fixed, independent of $\tilde{\lambda}_i^s$ or $\tilde{\lambda}_i^{s'}$. Suppose, for definiteness, that it's positive. In that case, if $\tilde{\lambda}_i^{s'}$ is larger for odour A than it is for odour B (at convergence), $\tilde{\lambda}_i^s$ will also be larger for odour A than for odour B. If, on the other hand, the

right hand side is negative, we'll see the opposite: $\tilde{\lambda}_i^s$ will be smaller for odour A than for odour B. Our model thus makes the prediction that if we plotted the values of two sisters cells at convergence for a range of odours, that plot would be monotonic.

For this prediction to hold, Eq (36) must be satisfied, which is guaranteed only when the periglomerular leak, $\varepsilon$, is zero. When $\varepsilon > 0$ on the other hand, Eq (33b) tells us that at convergence,

$$\lambda_i^s = \overline{\lambda}_i + \varepsilon \mu_i^s. \tag{38}$$

The ratio of sister mitral cell activations will thus acquire an odour dependence. But as we saw in Fig 10B, that dependence is relatively weak. Consistent with this, when we tested for monotonicity numerically, we found that it is largely maintained, as can be seen in Fig 13. We thus arrive at our main prediction, which holds even when there is periglomerular leak: at convergence, the activity of any mitral cell is an approximately monotonic function of the activity of any of its sisters.

## 3 Discussion

Most neural implementations of probabilistic inference require dense or all-to-all connectivity between elements, so that the explanatory power of all latent variables can be correctly accounted for. In common sensory settings where inference over hundreds of thousands of latent variables is not uncommon, such connectivity can require individual neurons to connect to hundreds of thousands of others, which is biologically implausible. In this work we have taken inspiration from the vertebrate olfactory system to show how such inference problems can be solved using sparse connectivity.

Naive olfactory inference would require each mitral cell to connect to hundreds of thousands of granule cells. However, in mice there are approximately 25–50 sister cells per glomerulus [22, 23], and we showed that the sisters can share the connectivity, resulting in a substantial reduction in the required number of synapses per mitral cell. However, this sharing of connectivity comes at a cost: it requires coordination by a neural population. We have identified that population as periglomerular cells, based on their pattern of connectivity.

Our approach is not limited to the particular inference setting presented here. It can be applied to the more complex generative model in [2], and a large class of nonlinear models (see Methods, Sec. 4.6). As another example, our approach can be readily applied to the 'sparse incomplete representations' of [36], whose Eqs. (5) and (6) are directly analogous to our Eq (10), and thus can be modified analogously to employ sparse connectivity.

### 3.1 Coordinating connectivity

One of the key requirements of our work is that sister cell connectivity matrices $W_{ij}^s$ sum according to Eq (14). This condition implies that the weights connecting a given granule cell $x_j$ to the sisters $\lambda_i^s$ in glomerulus $i$ sum to $A_{ij}$. In the sparsest connectivity setting this requires that each granule cell connect to exactly one sister cell from each glomerulus. This may seem difficult to implement, but given the similarity in the temporal responses of sister cells, such sparse connectivity may be achievable through lateral inhibition among granule cell spines, as two spines each contacting sisters from the same glomerulus would receive very similar inputs, motivating the elimination of one to reduce redundancy. Nevertheless, although sparsifying connectivity was the motivation for our model, the model does not require it, as long as the condition in Eq (14) is met.

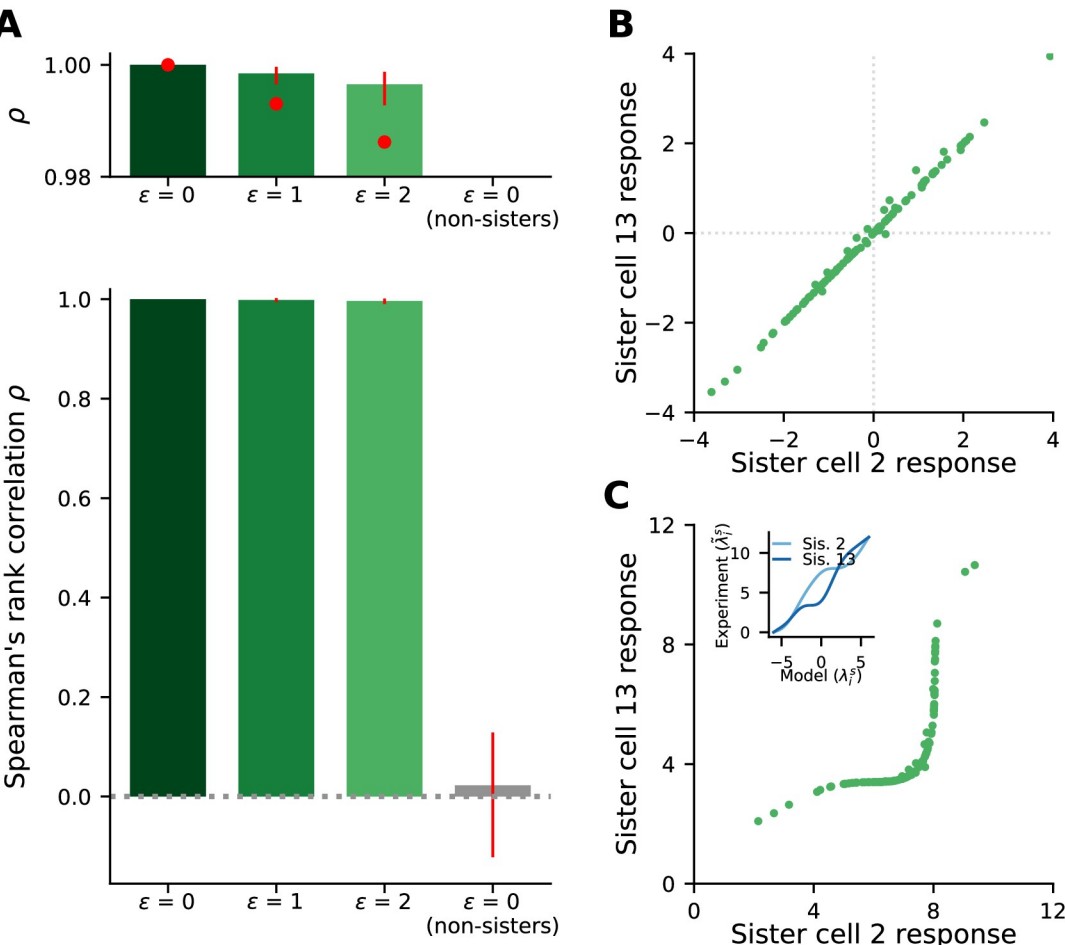

**Fig 13. Sister cell odour responses at convergence are approximately monotonically related, even when there is periglomerular leak.** (A) Full (bottom) and zoomed (top) distributions of Spearman's rank correlation, $\rho$, comparing responses at convergence for pairs of sister cells (colours) and non-sisters (gray) in a single model olfactory bulb. The olfactory bulb was stimulated with 100 random odours; data is from all 50 glomeruli. Bars indicate the median, red lines indicate the interquartile range, red dots indicate the minimum correlation observed. Olfactory bulb parameters were as in Table 1 except $S = 25$, and leak as indicated. (B) Responses at convergence of a pair of sister cells with the median Spearman's rank correlation in the highest leak setting, ($\varepsilon = 2$, top of right-most bar of panel A) to the 100 odours tested. (C) Responses from panel (B) after transformation through cell-specific monotonic nonlinearities (inset) modeling the transformation from model variables $\lambda_i^s$ to experimentally recorded values $\tilde{\lambda}_i^s$. Because the transformations are monotonic, the Spearman's rank correlations in panels B and C are identical.

Perhaps a more fundamental issue is that we have assumed throughout that the correct connectivity—the affinity matrix $A_{ij}$—is known. The key question is then how such 'correct connectivity' patterns can be learned, particularly in a setting where connectivity has to be coordinated across sister cells according to conditions like Eq (14). Previous work [37] and current investigations [38] have examined this issue. However, neither of those studies used sister mitral cells to sparsify connectivity, and how such circuitry could be learned remains an open question.

### 3.2 Cortical readout

Although our work deals mainly with olfactory computation in the bulb, communicating the results of this computation to the cortex is obviously required. In Methods, Sec. 4.5, we outline

one scheme by which this could be done, in which cortical neurons effectively mirror the computation of olfactory bulb granule cells. If the projections from the mitral cells to the piriform cortex satisfy Eq (95), the cortical readout at convergence would be an exact copy of the granule cell activations in the bulb. Needless to say, the main problem with such a scheme is how to satisfy this condition on the weights, and is likely made more difficult by the physical separation of the bulb and the cortex. One possibility is that the feedback connections from the cortex to the bulb could be used to learn the right connectivity. Another possibility is that an exact copy of the granule cell output is not required, and that random projections from the bulb to the cortex would suffice [39]. We leave the resolution of such issues to future work.

### 3.3 Predictions

Our model makes a number of experimentally testable predictions. Perhaps the easiest to test is the monotonicity condition in Eq (37), supported by the numerical results in Fig 13, stating that the activity of a mitral cell at convergence is an approximately monotonic function of the activity of any of its sisters. It is admittedly hard to define convergence in animals that are subject to periodic olfactory input due to the breathing cycle. Nevertheless, we can approximate it by the activity at the end of each breathing cycle in an anesthetized animal presented with an odour, after several breathing cycles have elapsed. Our model would then predict that for any pair of sister cells, plotting their firing rates at convergence in response to a range of odours would reveal an approximately monotonic function.

A key computation required by our model is the coordination of sister mitral cell activity to arrive at the MAP solution, which we propose is performed by the periglomerular cells. Therefore, we predict that deactivating the periglomerular cells should eliminate sister cell coordination and push the results of inference away from the MAP solution, and likely produce widespread, low amplitude activation of granule cells (see Methods, Sec. 4.3).

Although maximum sparsity is not required of our model (see, e.g., Fig 3B), if, as we propose, the function of the sister mitral cells is to sparsify the mitral-granule cell connectivity required for inference, then maximally sparse connectivity solutions would be expected, in which granule cells contact at most one sister cell per glomerulus. Connectomics studies in the olfactory bulb should be able to tell us how often granule cells receive input from two or more sister mitral cells.

Finally, in our model we grouped all olfactory bulb projection neurons together and referred to them as mitral cells. This was for simplicity only. In fact, though, olfactory bulb projection neurons fall into two anatomically and physiologically distinct classes, mitral and tufted cells, and it is likely that these classes play different roles in olfactory processing [19]. Because our model describes how a given computation can be distributed among sister cells, if mitral and tufted cells perform different computations, then our predictions would apply independently to sister mitral cells and to sister tufted cells. It would, therefore, be important to establish that two sisters are of the same type before applying our prediction of sister cell monotonicity.

### 3.4 Summary

In summary, taking inspiration from the sister mitral cells in the vertebrate olfactory bulb, we showed how inference that typically requires dense connectivity can be performed using sparse connectivity. This means computations that would normally require hundreds of thousands of connections can be performed with a fraction of that. To the best of our knowledge our work is the first to propose a computational role for the sister mitral cells, and it makes a number of

experimentally testable predictions. Despite its olfactory origins, our approach is quite general, and can be applied in other inference models.

# 4 Methods

Here we provide additional details about the analyses used in the Main Text. In Sec. 4.1 we introduce a Lagrangian for the non-leaky ($\varepsilon = 0$) system, and modify it for the leaky case to gain insight into the resulting dynamics. In Sec. 4.2 we perform linear analysis to determine how the various parameters influence the transient response properties of the system. In Sec. 4.3 we examine the implications of eliminating the periglomerular cells. In Sec. 4.4 we provide additional details about our simulations of the system dynamics. In Sec. 4.5 we propose a method for reading out odour concentrations in cortex. Finally, in Sec. 4.6 we show that sister mitral cells can be applied to other inference models.

## 4.1 Leaky periglomerular cells

To gain insight into the effect of pergilomerular leak, we start with a Lagrangian for the non-leaky setting,

$$\mathcal{L}(\mathbf{x}, \boldsymbol{\mu}, \boldsymbol{\lambda}) = \sum_j \phi(x_j) + \sum_{i,s} \left[ \frac{\lambda_i^s}{S} \left( y_i - S \sum_j W_{ij}^s x_j \right) - \frac{\sigma^2 (\lambda_i^s)^2}{2S} - \mu_i^s (\lambda_i^s - \overline{\lambda}_i) \right]. \tag{39}$$

It is straightforward to show that

$$\max_{\boldsymbol{\lambda}} \min_{\boldsymbol{\mu}} [\mathcal{L}(\mathbf{x}, \boldsymbol{\mu}, \boldsymbol{\lambda})] = \sum_{j=1}^N \phi(x_j) + \frac{1}{2\sigma^2} \sum_{i=1}^M \left( y_i - \sum_{j=1}^N A_{ij} x_j \right)^2. \tag{40}$$

Consequently, if we minimize the Lagrangian with respect to $\mathbf{x}$ and $\boldsymbol{\mu}$ and maximize it with respect to $\boldsymbol{\lambda}$, we recover the MAP solution. The resulting equations are

$$\frac{dx_j}{dt} \propto -\frac{\partial \mathcal{L}(\mathbf{x}, \boldsymbol{\mu}, \boldsymbol{\lambda})}{\partial x_j} = -\frac{\partial \phi}{\partial x_j} + \sum_{i,s} W_{ij}^s \lambda_i^s. \tag{41a}$$

$$\frac{d\lambda_i^s}{dt} \propto \frac{\partial \mathcal{L}(\mathbf{x}, \boldsymbol{\mu}, \boldsymbol{\lambda})}{\partial \lambda_i^s} = \frac{\sigma^2}{S} \left[ -\lambda_i^s + \frac{1}{\sigma^2} \left( y_i - S \sum_j W_{ij}^s x_j - S(\mu_i^s - \overline{\mu}_i) \right) \right] \tag{41b}$$

$$\frac{d\mu_i^s}{dt} \propto -\frac{\partial \mathcal{L}(\mathbf{x}, \boldsymbol{\mu}, \boldsymbol{\lambda})}{\partial \mu_i^s} = \lambda_i^s - \overline{\lambda}_i. \tag{41c}$$

These equations correspond, up to scaling factors, to Eqs (23), (20) and (19), respectively, except that the second equation above includes the additional term $S\overline{\mu}_i$. That term is, in fact, necessary to yield the correct dynamics. We dropped it because we can instead choose the initial conditions so that $\overline{\mu}_i$ is zero, and under the periglomerular dynamics it will stay zero forever. Thus the circuit dynamics can be viewed as finding the saddle point of the Lagrangian of a constrained optimization problem derived from the original MAP objective.

To gain insight into the effect of the periglomerular leak we add a term proportional to $(\mu_i^s)^2$ to the Lagrangian in Eq (39),

$$\mathcal{L}_\varepsilon(\mathbf{x}, \boldsymbol{\lambda}, \boldsymbol{\mu}) = \sum_j \phi(x_j) + \sum_{i,s} \left[ \lambda_i^s \left( \frac{y_i}{S} - \sum_j W_{ij}^s x_j \right) - \frac{\sigma^2 (\lambda_i^s)^2}{2S} - \mu_i^s (\lambda_i^s - \overline{\lambda}_i) + \frac{\varepsilon}{2} (\mu_i^s)^2 \right]. \tag{42}$$

As is not hard to show, this introduces a term $-\varepsilon\mu_i^s$ to the right hand side of Eq (41c), which is the same as the leak term in Eq (33b). To give us a Lagrangian that depends solely on $\mathbf{x}$, we eliminate the auxiliary variables $\mu_i^s$ and $\lambda_i^s$. We start by minimizing $\mathcal{L}_\varepsilon(\mathbf{x}, \boldsymbol{\lambda}, \boldsymbol{\mu})$ with respect to $\mu_i^s$, yielding

$$\mu_i^s = \frac{1}{\varepsilon}\left(\lambda_i^s - \overline{\lambda}_i\right), \tag{43}$$

so that

$$\mathcal{L}_\varepsilon(\mathbf{x}, \boldsymbol{\lambda}) = \sum_j \phi(x_j) + \sum_{i,s}\left[\lambda_i^s\left(\frac{y_i}{S} - \sum_j W_{ij}^s x_j\right) - \frac{\sigma^2(\lambda_i^s)^2}{2S} - \frac{(\lambda_i^s - \overline{\lambda}_i)^2}{2\varepsilon}\right]. \tag{44}$$

The final term couples the $\lambda_i^s$ to their mean value computed over sisters. Extremizing with respect to $\lambda_i^s$ yields

$$\lambda_i^s = \frac{1}{\sigma^2}\left(y_i - q_\varepsilon\sum_j A_{ij}x_j - (1 - q_\varepsilon)\sum_j SW_{ij}^s x_j\right) \tag{45}$$

where

$$q_\varepsilon \equiv \frac{S}{S + \varepsilon\sigma^2} \tag{46}$$

and we used, as usual, the constraint on the weights, Eq (14). We can now insert this into Eq (44) to derive a Lagrangian that depends only on $\mathbf{x}$. Doing that is straightforward, although somewhat algebra-intensive, but it ultimately yields Eq (26).

## 4.2 Linear dynamics

In addition to the number of sister cells $S$, our model has several parameters that affect the transient dynamics. To understand these effects, we performed linear analysis of the model in Eq (33). Our aim is a qualitative understanding, so we frequently opt for approximations that yield simple and tractable results over exact solutions.

We perform the linear analysis around the steady-state solutions to Eq (33). Because of the threshold nonlinearity in Eq (33d) only a small number of granule cell activations $x_i$ will be non-zero. We take the deviations around steady-state small enough so that the composition of this active set does not change. Adopting notation where vectors are in sister cell space, we write

$$\boldsymbol{\lambda}_i = \boldsymbol{\lambda}_{0i} + \delta\boldsymbol{\lambda}_i \tag{47a}$$

$$\boldsymbol{\mu}_i = \boldsymbol{\mu}_{0i} + \delta\boldsymbol{\mu}_i \tag{47b}$$

$$x_i = x_{0i} + \delta x_i \tag{47c}$$

$$v_i = v_{0i} + \delta v_i. \tag{47d}$$

Here quantities with the subscript 0 are steady-state solutions to Eq (33); quantities with a $\delta$ in

front of them are infinitesimally small, and obey linear dynamics

$$\tau_\lambda \frac{d\delta\boldsymbol{\lambda}_i}{dt} = -\delta\boldsymbol{\lambda}_i - S\eta \sum_{j=1}^{n} \mathbf{w}_{ij}\delta v_j - \gamma S\eta \delta\boldsymbol{\mu}_i \tag{48a}$$

$$\tau_\mu \frac{d\delta\boldsymbol{\mu}_i}{dt} = -\boldsymbol{\varepsilon}\delta\boldsymbol{\mu}_i + \delta\boldsymbol{\lambda}_i - \frac{\mathbf{1}\mathbf{1}\cdot\delta\boldsymbol{\lambda}_i}{S} \tag{48b}$$

$$\tau_v \frac{d\delta v_j}{dt} = -\delta v_j + \sum_{i=1}^{M} \delta\boldsymbol{\lambda}_i \cdot \mathbf{w}_{ij} \tag{48c}$$

where the $s^{\text{th}}$ component of $\mathbf{w}_{ij}$ is $W_{ij}^s$ and

$$\eta \equiv \frac{1}{\gamma\sigma^2} . \tag{49}$$

Because these are linear equations, they (generically) have solutions whose temporal part is given by $e^{\xi t}$. Consequently, derivatives with respect to time can be replaced by $\xi$, leading to

$$(\tau_\lambda \xi + 1)\,\delta\boldsymbol{\lambda}_i = -S\eta \sum_{j=1}^{n} \mathbf{w}_{ij}\delta v_j - \gamma\eta S\delta\boldsymbol{\mu}_i \tag{50a}$$

$$(\tau_\mu \xi + \boldsymbol{\varepsilon})\,\delta\boldsymbol{\mu}_i = \delta\boldsymbol{\lambda}_i - \frac{\mathbf{1}\mathbf{1}\cdot\delta\boldsymbol{\lambda}_i}{S} \tag{50b}$$

$$(\tau_v \xi + 1)\,\delta v_j = \sum_{i=1}^{M} \delta\boldsymbol{\lambda}_i \cdot \mathbf{w}_{ij} . \tag{50c}$$

Our approach is to transform this set of equations to an eigenvalue equation in a single variable. To that end we eliminate $\delta\boldsymbol{\mu}_i$ and $\delta v_j$, leaving us, after a small amount of algebra, with

$$(\tau_\mu \xi + \boldsymbol{\varepsilon})(\tau_\lambda \xi + 1)\,\delta\boldsymbol{\lambda}_i + \gamma\eta(S\,\delta\boldsymbol{\lambda}_i - \mathbf{1}\mathbf{1}\cdot\delta\boldsymbol{\lambda}_i) = -\eta S \frac{\tau_\mu \xi + \boldsymbol{\varepsilon}}{\tau_v \xi + 1} \sum_{k=1}^{M} \left( \sum_{j=1}^{n} \mathbf{w}_{ij}\mathbf{w}_{kj} \right) \cdot \delta\boldsymbol{\lambda}_k . \tag{51}$$

The $S \times S$ term in parentheses on the right hand side is the $(i, k)^{\text{th}}$ block of an $MS \times MS$ matrix of rank $n$, whereas the set of vectors $\delta\boldsymbol{\lambda}_k$ contain $MS$ components ($k$ runs from 1 to $M$ and $\delta\boldsymbol{\lambda}_k$ is an $S$-dimensional vector). Consequently, so long as $n < MS$ (the regime we consider), that rank $n$ matrix has two different classes of eigenvectors: those with zero eigenvalue and those with non-zero eigenvalue. We consider the former first.

For eigenvectors with zero eigenvalue, the left hand side of Eq (51) must be zero. For that to happen there are, naively, two possibilities: $\delta\boldsymbol{\lambda}_i \propto \mathbf{1}$, in which case $\xi$ obeys

$$(\tau_\mu \xi + \boldsymbol{\varepsilon})(\tau_\lambda \xi + 1) = 0 , \tag{52}$$

and $\delta\boldsymbol{\lambda}_i \cdot \mathbf{1} = 0$, in which case $\xi$ obeys

$$(\tau_\mu \xi + \boldsymbol{\varepsilon})(\tau_\lambda \xi + 1) + \gamma\eta S = 0 . \tag{53}$$

Again naively, this should result in four eigenvalues so long as $MS > n$.

To make this rigorous—and to uncover exactly when the above eigenvalues apply (as the naive conclusions are not quite correct)—we need a more involved analysis. Before proceeding with the general case, however, we note that there's a special case: $\tau_\mu \xi + \boldsymbol{\varepsilon} = 0$, since in that case

the right hand side of Eq (51) is identically zero. Examining Eq (50b) we see that when this holds, $\delta\boldsymbol{\lambda}_i \propto \mathbf{1}$, which then determines $\delta v_j$ by Eq (50c), and $\delta\boldsymbol{\mu}_i$ by Eq (50b). Thus there are $M$ modes, corresponding to the root of Eq (52); for these,

$$\xi = -\frac{\varepsilon}{\tau_\mu} \ . \tag{54}$$

When $\tau_\mu\xi + \varepsilon \neq 0$, there is an $(MS - n)$-dimensional space of vectors, denoted $\delta\boldsymbol{\lambda}_k^\mu$, that obey

$$\sum_{k=1}^{M}\left(\sum_{j=1}^{n}\mathbf{w}_{ij}\mathbf{w}_{kj}\right)\cdot\delta\boldsymbol{\lambda}_k^\mu = 0\,. \tag{55}$$

We need to choose a linear combination of these for which the left hand side of Eq (51) is zero; that is, we need to choose a set of $a_\mu$ such that (after rearranging terms slightly)

$$0 = \frac{(\tau_\mu\xi + \varepsilon)(\tau_\lambda\xi + 1)}{S}\sum_{\mu=1}^{MS-n}a_\mu\mathbf{1}\mathbf{1}\cdot\delta\boldsymbol{\lambda}_i^\mu$$

$$+ \left[(\tau_\mu\xi + \varepsilon)(\tau_\lambda\xi + 1) + \gamma\eta S\right]\sum_{\mu=1}^{MS-n}a_\mu\left(\mathbf{I} - \frac{\mathbf{1}\mathbf{1}}{S}\right)\cdot\delta\boldsymbol{\lambda}_i^\mu\,. \tag{56}$$

Since this equation must be satisfied for all $i$, there are $MS$ equations ($i$ runs form 1 to $M$ and $\delta\boldsymbol{\lambda}_i^\mu$ is an $S$-dimensional vector). But there are only $MS - n$ adjustable parameters, so in general the only solution has all the $a_\mu = 0$. There are, though, two ways to find nontrivial solutions. One is to set the first term in parentheses to zero (i.e., enforce Eq (52)). Then, because $(\mathbf{I} - \mathbf{1}\mathbf{1}/S)\cdot\delta\boldsymbol{\lambda}_i^\mu$ spans $S - 1$ dimensions, Eq (56) corresponds to $M(S - 1)$ equations. Because there are $MS - n$ adjustable parameters, there is a nontrivial solution if $MS - n - M(S - 1) > 0$; that is, if $M > n$. Note that because we have already taken into account the solution with $\tau_\mu\xi + \varepsilon = 0$, these solutions must have $\tau_\lambda\xi + 1 = 0$. The other way to find nontrivial solutions is to set the term in brackets in Eq (56) to zero (i.e., enforce Eq (53)). Then, because $\mathbf{1}\mathbf{1}\cdot\delta\boldsymbol{\lambda}_i^\mu$ spans one dimension, Eq (56) corresponds to $M$ equations, and so there is a nontrivial solution if $MS - n - M > 0$; $MS - n - M > 0$; that is, $M(S - 1) > n$.

In summary, when the right hand side of Eq (51) is zero, we have the following eigenvalues, all of which are exact: $M$ modes with

$$\xi = -\frac{\varepsilon}{\tau_\mu}\,, \tag{57}$$

$[M - n]^+$ modes with

$$\xi = -\frac{1}{\tau_\lambda}\,, \tag{58}$$

and $[M(S - 1) - n]^+$ modes with

$$\xi^{\mathrm{mp}} = \frac{-(\tau_\mu + \varepsilon\tau_\lambda)}{2\tau_\mu\tau_\lambda} \pm i\sqrt{\frac{\gamma\eta S}{\tau_\mu\tau_\lambda} - \frac{(\tau_\mu - \varepsilon\tau_\lambda)^2}{4\tau_\mu^2\tau_\lambda^2}} \tag{59}$$

where the superscript 'mp' indicates that this mode involves only mitral and periglomerular cells (see next paragraph), and $i = \sqrt{-1}$ (when it's is not an index). Because $\xi^{\mathrm{mp}}$ can take on two values, there are $2[M(S - 1) - n]^+$ modes of this type.

Two comments are in order. First, when there is only one sister cell ($S = 1$), the mode in Eq (59) does not exist, as that mode requires $M(S - 1) > n$. Second, for the modes given in Eqs (57), (58) and (59), $\sum_k \mathbf{w}_{kj} \cdot \delta\boldsymbol{\lambda}_k = 0$; this in turn implies, via Eq (50c), that $\delta v_j = 0$. Thus, these modes involve the periglomerular and mitral cells, but not the granule cells.

When the right hand side of Eq (51) is nonzero, analysis in $\delta\lambda_i$ space is difficult. However, we can instead work in $\delta v_j$ space: eliminating $\delta\lambda_i$ and $\delta\boldsymbol{\mu}_i$ from Eq (50), we can write down an eigenvalue equation for $\delta v_j$; after tedious but straightforward algebra, including application of the Sherman-Morrison formula to invert the operator on the left-hand side of Eq (51), we arrive at

$$
(\tau_v \xi + 1)\delta v_i = -\frac{\eta S(\tau_\mu \xi + \boldsymbol{\varepsilon})}{(\tau_\mu \xi + \boldsymbol{\varepsilon})(\tau_\lambda \xi + 1) + \gamma\eta S}
$$
$$
\times \sum_{j=1}^{n}\left(\frac{\eta\gamma}{(\tau_\mu \xi + \boldsymbol{\varepsilon})(\tau_\lambda \xi + 1)}\sum_{k=1}^{M}A_{ki}A_{kj} + \sum_{k=1}^{M}\mathbf{w}_{ki} \cdot \mathbf{w}_{kj}\right)\delta v_j
\tag{60}
$$

where we used Eq (14) to write $\mathbf{1} \cdot \mathbf{w}_{kj} = A_{kj}$.

Finding exact non-trivial solutions to Eq (60) requires finding the eigenvalues of the sum of the two matrices on the right hand side of this equation. That's difficult in general, so instead we make an approximation: we replace the second matrix with the identity. We justify this by arguing that its eigenvalues are narrowly distributed around 1. To show that, we start by writing

$$
\sum_k \mathbf{w}_{ki} \cdot \mathbf{w}_{kj} = \sum_{k=1}^{M}\sum_{s=1}^{S} W_{ki}^s W_{kj}^s.
\tag{61}
$$

For a given $k$ and $j$, The elements of $W_{kj}^s$ are nonzero for only one value of $s$ and zero for the rest (see Eq (18)). Consequently, they are not *iid*, which makes it difficult to compute the eigenvalue spectrum. However, a reasonable approximation to $W_{kj}^s$ is

$$
W_{kj}^s = \begin{cases} A_{kj} & \text{probability } 1/S \\ 0 & \text{probability } 1 - 1/S. \end{cases}
\tag{62}
$$

In that case, $\text{var}(W_{kj}^s) = \text{var}(A_{kj})/S = 1/MS$, implying that $\sum_k \mathbf{w}_{ki} \cdot \mathbf{w}_{kj}$ approximately follows a Marcenko-Pastur distribution with parameters $(1, n/MS)$ [40]. For this distribution, the eigenvalues lie in the range

$$
\left(1 - \sqrt{n/MS}\right)^2 \leq \text{eigenvalues} \leq \left(1 + \sqrt{n/MS}\right)^2.
\tag{63}
$$

When $n \ll MS$, the regime of interest, these eigenvalues are very narrowly distributed around 1. Thus, the matrix in Eq (61) is, to good approximation, the identity. This approximation breaks down as $n$ increases, but we're mainly interested in small $n$, so that is not a problem.

With this approximation, the only nontrivial matrix left in Eq (60) is the one involving the $A_{kj}$. The elements of $A_{kj}$ are drawn *iid* from $\mathcal{N}(0, 1/M)$, so

$$
\sum_{k=1}^{M}A_{ki}A_{kj} \sim \text{MP}\left(1, \frac{n}{M}\right)
\tag{64}
$$

where MP denotes the Marcenko-Pastur distribution. Using $v$ to denote an eigenvalue of this

distribution, we see that Eq (60) can be approximated as

$$
(\tau_\nu \xi + 1) \approx -\frac{\eta S(\tau_\mu \xi + \varepsilon)}{(\tau_\mu \xi + \varepsilon)(\tau_\lambda \xi + 1) + \gamma \eta S} \left(1 + \frac{\nu \eta \gamma}{(\tau_\mu \xi + \varepsilon)(\tau_\lambda \xi + 1)}\right).
\tag{65}
$$

There are $n$ eigenvalues, corresponding to the fact that $j$ runs from 1 to $n$ in Eq (60), so there are $n$ sets of solutions to this equation. (We say "sets of solutions", rather than just one, because Eq (65) is a polynomial in $\xi$, which has several roots). The positive eigenvalues lie in the range

$$
(1 - \sqrt{n/M})^2 < \nu < (1 + \sqrt{n/M})^2.
\tag{66}
$$

If $n < M$, all of the eigenvalues lie in this range, while if $n \geq M$, only $M$ eigenvalues lie in this range; the other $n - M$ are zero.

**4.2.1 Approximate solutions.** To solve to Eq (65), our first step is to write it is a polynomial,

$$
\begin{aligned}
q(\xi) &= \tau_\lambda^2 \tau_\mu \tau_\nu \xi^4 + (\tau_\lambda^2 \tau_\mu + 2\tau_\lambda \tau_\mu \tau_\nu + \varepsilon \tau_\lambda^2 \tau_\nu)\xi^3 \\
&\quad + (\gamma \eta S \tau_\lambda \tau_\nu + \eta S \tau_\lambda \tau_\mu + 2\tau_\lambda \tau_\mu + \tau_\mu \tau_\nu + \varepsilon \tau_\lambda^2 + 2\varepsilon \tau_\lambda \tau_\nu)\xi^2 \\
&\quad + (\gamma \eta S \tau_\lambda + \gamma \eta S \tau_\nu + \eta S \tau_\mu + \eta S \varepsilon \tau_\lambda + \tau_\mu + \varepsilon \tau_\nu + 2\varepsilon \tau_\lambda)\xi \\
&\quad + \nu \gamma \eta^2 S + \gamma \eta S + \varepsilon \eta S + \varepsilon.
\end{aligned}
\tag{67}
$$

We're looking for values of $\xi$ that satisfy $q(\xi) = 0$. Note that $q(\xi)$ depends on $\nu$, which means solutions to $q(\xi) = 0$ will also depend on $\nu$; we drop that dependence to reduce clutter.

Because $q(\xi)$ is quartic, an exact analytic expression for its roots is available, but it is too complex to yield insight. Instead, we take a perturbative approach, which rests on the observation that $\eta$ is large, on the order of 100 (see its definition, Eq (49) and Table 1). To take advantage of this, we scale $\xi$ by a factor of $\sqrt{\eta}$. Choosing a scaling that gives us a dimensionless quantity, we make the change of variables

$$
\xi = \sqrt{\eta S} \frac{\alpha}{\tau_\lambda}.
\tag{68}
$$

Then, defining the time constant ratios

$$
\kappa_\mu \equiv \frac{\tau_\lambda}{\tau_\mu}
\tag{69a}
$$

$$
\kappa_\nu \equiv \frac{\tau_\lambda}{\tau_\nu},
\tag{69b}
$$

and working to first order in $1/\sqrt{\eta S}$ we find, after straightforward algebra, that $q(\xi)$, expressed in terms of $\alpha$ (and denoted, in a slight abuse of notation, $q(\alpha)$) is given approximately by

$$
q(\alpha) \dot{\approx} B(\alpha) + \frac{b(\alpha)}{\sqrt{\eta S}},
\tag{70}
$$

where $\dot{\approx}$ indicates approximate equality up to multiplicative constant and

$$
B(\alpha) \equiv \alpha^4 + (\kappa_\nu + \gamma \kappa_\mu)\alpha^2 + \nu \kappa_\mu \kappa_\nu \frac{\gamma}{S}
\tag{71a}
$$

$$
b(\alpha) \equiv (2 + \kappa_\nu + \varepsilon \kappa_\mu)\alpha^3 + (\kappa_\nu + \gamma \kappa_\mu + (\gamma + \varepsilon)\kappa_\mu \kappa_\nu)\alpha.
\tag{71b}
$$

In the large $\eta$ limit, the roots of $q(\alpha)$ are determined by those of $B(\alpha)$. Defining

$$\alpha_{\pm}^2 \equiv \frac{-(\kappa_{\nu} + \gamma\kappa_{\mu}) \pm \sqrt{(\kappa_{\nu} + \gamma\kappa_{\mu})^2 - 4\nu\gamma\kappa_{\nu}\kappa_{\mu}/S}}{2} ,\tag{72}$$

the corresponding four roots are $\pm i\alpha_{\pm}$. The argument of the square root is $(\kappa_{\nu} - \gamma\kappa_{\mu})^2 + 4\gamma\kappa_{\nu}$ $\kappa_{\mu}(1 - \nu/S)$, which is guaranteed to be positive if $\nu < S$. From Eq (66), this requires $n/MS < (1 - 1/\sqrt{S})^2$. We'll restrict ourselves to this regime, which ensures that both $\alpha_+^2$ and $\alpha_-^2$ are negative, which in turn means all four of these roots are purely imaginary.

To compute the corrections to these roots, we let $\alpha = \alpha_0 + \alpha_1$ where $\alpha_0$ is any of the above four roots. Then, performing a Taylor expansion of $q(\alpha)$, Eq (70), around $\alpha_0$, we have

$$q(\alpha_0 + \alpha_1) \doteq B(\alpha_0) + \frac{b(\alpha_0)}{\sqrt{\eta S}} + B'(\alpha_0)\alpha_1 .\tag{73}$$

Setting this to zero and solving for $\alpha_1$ gives

$$\alpha_1 \approx -\frac{1}{\sqrt{\eta S}}\frac{b(\alpha_0)}{B'(\alpha_0)} .\tag{74}$$

Using Eq (71) for $B(\alpha_0)$ and $b(\alpha_0)$, setting $\alpha_0^2$ to $\alpha_{\pm}^2$, and using Eq (72) to simplify the resulting expression, we arrive at

$$\alpha_{1,\pm} = -\frac{1}{2\sqrt{\eta S}}\frac{(2 + \kappa_{\nu} + \varepsilon\kappa_{\mu})\alpha_{\pm}^2 + \kappa_{\nu} + \gamma\kappa_{\mu} + (\gamma + \varepsilon)\kappa_{\mu}\kappa_{\nu}}{\pm\sqrt{(\kappa_{\nu} + \gamma\kappa_{\mu})^2 - 4\nu\gamma\kappa_{\nu}\kappa_{\mu}/S}} .\tag{75}$$

We thus have (using Eq (68))

$$\zeta_{\pm}^{\text{high}} = \frac{\sqrt{\eta S}}{\tau_{\lambda}}\left[\alpha_{1,-} \pm i\sqrt{-\alpha_-^2}\right]\tag{76a}$$

$$\zeta_{\pm}^{\text{low}} = \frac{\sqrt{\eta S}}{\tau_{\lambda}}\left[\alpha_{1,+} \pm i\sqrt{-\alpha_+^2}\right]\tag{76b}$$

with $\alpha_{1,\pm}$ given in Eq (68) and $\alpha_{\pm}^2$ given in Eq (72). The "high" and "low" superscripts refer to the fact that $|\alpha_-^2| > |\alpha_+^2|$, as is easy to see from Eq (72).

It is instructive to consider the large $S$ limit, which greatly simplifies the roots. Focusing first on the high frequency roots, in this limit we have

$$\alpha_-^2 \approx -(\kappa_{\nu} + \gamma\kappa_{\mu})\tag{77a}$$

$$\alpha_{1,-} \approx -\frac{1}{2\sqrt{\eta S}}\frac{\kappa_{\nu} + \gamma\kappa_{\mu} + \kappa_{\nu}^2 + \gamma\varepsilon\kappa_{\mu}^2}{\kappa_{\nu} + \gamma\kappa_{\mu}}\tag{77b}$$

so that, using Eq (68),

$$\zeta_{\pm}^{\text{high}} \approx -\frac{1}{2}\left(\frac{1}{\tau_{\nu}} + \frac{1}{\tau_{\lambda}} + \frac{\varepsilon}{\tau_{\mu}} - \frac{\gamma + \varepsilon}{\gamma\tau_{\nu} + \tau_{\mu}}\right) \pm i\sqrt{\frac{S\eta}{\tau_{\lambda}}\left(\frac{1}{\tau_{\nu}} + \frac{\gamma}{\tau_{\mu}}\right)} .\tag{78}$$

To approximate the low frequency roots, which correspond to $\alpha_+^2$, we perform a Taylor

expansion of the square root in Eq (72) around $(\kappa_\nu + \gamma\kappa_\mu)^2$, yielding

$$\alpha_+^2 \approx -\frac{\nu\gamma\kappa_\nu\kappa_\mu}{S(\kappa_\nu + \gamma\kappa_\mu)} \tag{79a}$$

$$\alpha_{1,+} \approx -\frac{1}{2\sqrt{\eta S}}\frac{\kappa_\nu + \gamma\kappa_\mu + (\gamma + \varepsilon)\kappa_\mu\kappa_\nu}{\kappa_\nu + \gamma\kappa_\mu} \tag{79b}$$

so that, using Eq (68),

$$\xi_\pm^{\text{low}} \approx -\frac{1}{2}\left(\frac{1}{\tau_\lambda} + \frac{\gamma + \varepsilon}{\gamma\tau_\nu + \tau_\mu}\right) \pm i\sqrt{\frac{1}{\tau_\lambda}\frac{\nu\eta\gamma}{\gamma\tau_\nu + \tau_\mu}}. \tag{80}$$

In summary, our goal was to find solutions to Eq (67) for each value of $\nu$, where $\nu$ is drawn from the Marchenko-Pastur distribution MP(1, $n/M$). This implies that there is a distribution of solutions, $\xi$, which we can find by solving for $\xi$ at each $\nu$. We did that perturbatively, yielding the high and low frequency solutions given in Eq (76) (with approximate expression for these quantities given in Eqs (78) and (80)). Note that if $\nu = 0$ (which can happen when $n > M$), $\alpha_+ = 0$ (see Eq (72)). When that happens, $\xi_\pm^{\text{low}}$ takes on only one value, not two (see Eq (76b)), and so there are three possible solutions.

The number of modes when the right hand side of Eq (51) is nonzero, then, depends on $n$. There are always $2n$ high frequency modes. When $n \leq M$ (so that $\nu$ is strictly positive), there are also $2n$ low-frequency oscillatory modes. When $n > M$, on the other hand, $n - M$ of the eigenvalues, $\nu$, are zero, and the rest are positive, resulting in $n - M$ decaying modes and $2M$ low-frequency oscillatory modes. We thus have $2n - [n - M]^+$ decaying and low-frequency oscillatory modes, for a total of

$$\text{number of modes} = 4n - [n - M]^+ \tag{81}$$

for Eq (60).

All modes of the system are tabulated in Table 2. They are given exactly by Eqs (57), (58) and (59), and approximately by (76). All of these modes have a decay associated with them, and the latter three also have oscillation frequencies. For simplicity, we considered the large $S$ limit, so we used Eqs (78) and (80) for the approximate modes given in Eq (76). Assuming $M(S - 1) > n$, the total number of modes is $M + [M - n]^+ + 2(M(S - 1) - n) + 4n - [n - M]^+$. Adding these together gives $2MS + n$, as it should.

**4.2.2 Stability.** The perturbative corrections in Eq (75) allow us to assess the stability of the linearized dynamics. For stability, both $\alpha_{1,+}$ and $\alpha_{1,-}$ (which are real) must be negative.

**Table 2. Linear analysis modes and their properties.** The last two modes correspond to the large $S$ limit of Eq (76).

| $\|\text{Re}(\xi)\|^{-1}$: Decay time constant | $\|\text{Im}(\xi)\|$: Oscillation frequency | Number of modes | Source |
|---|---|---|---|
| $\dfrac{\tau_\lambda}{\varepsilon\kappa_\mu}$ | 0 | $M$ | Eq (57) |
| $\tau_\lambda$ | 0 | $[M - n]^+$ | Eq (58) |
| $\dfrac{2\tau_\lambda}{1 + \varepsilon\kappa_\mu}$ | $\dfrac{1}{\tau_\lambda}\sqrt{\gamma\eta\kappa_\mu S}$ | $2[M(S - 1) - n]^+$ | Eq (59) |
| $\dfrac{2(\gamma\kappa_\mu + \kappa_\nu)\tau_\lambda}{\kappa_\nu^2 + \gamma\varepsilon\kappa_\mu^2 + \gamma\kappa_\mu + \kappa_\nu}$ | $\dfrac{1}{\tau_\lambda}\sqrt{\eta(\gamma\kappa_\mu + \kappa_\nu)S}$ | $2n$ | Eq (78) |
| $\dfrac{2(\gamma\kappa_\mu + \kappa_\nu)\tau_\lambda}{(\gamma + \varepsilon)\kappa_\mu\kappa_\nu + \gamma\kappa_\mu + \kappa_\nu}$ | $\dfrac{1}{\tau_\lambda}\sqrt{\dfrac{\nu\eta\gamma\kappa_\nu\kappa_\nu}{\gamma\kappa_\mu + \kappa_\nu}}$ | $2n - [n - M]^+$ | Eq (80) |

Combining Eq (72) with Eq (75), we see that this gives us the two conditions,

$$-(\kappa_\nu + \gamma\kappa_\mu) + \sqrt{(\kappa_\nu + \gamma\kappa_\mu)^2 - 4\nu\gamma\kappa_\nu\kappa_\mu/S} + \frac{2(\kappa_\nu + \gamma\kappa_\mu + (\gamma + \varepsilon)\kappa_\mu\kappa_\nu)}{2 + \kappa_\nu + \varepsilon\kappa_\mu} > 0 \tag{82a}$$

$$-(\kappa_\nu + \gamma\kappa_\mu) - \sqrt{(\kappa_\nu + \gamma\kappa_\mu)^2 - 4\nu\gamma\kappa_\nu\kappa_\mu/S} + \frac{2(\kappa_\nu + \gamma\kappa_\mu + (\gamma + \varepsilon)\kappa_\mu\kappa_\nu)}{2 + \kappa_\nu + \varepsilon\kappa_\mu} < 0, \tag{82b}$$

which can be simplified to just one,

$$\sqrt{(\kappa_\nu + \gamma\kappa_\mu)^2 - 4\nu\gamma\kappa_\nu\kappa_\mu/S} > \left| \kappa_\nu + \gamma\kappa_\mu - \frac{2(\kappa_\nu + \gamma\kappa_\mu + (\gamma + \varepsilon)\kappa_\mu\kappa_\nu)}{2 + \kappa_\nu + \varepsilon\kappa_\mu} \right|$$
$$= \frac{|\kappa_\nu - \varepsilon\kappa_\mu|}{\kappa_\nu + \varepsilon\kappa_\mu + 2} |\kappa_\nu - \gamma\kappa_\mu|. \tag{83}$$

As above (see comments following Eq (72)), in the regime of interest, $n/MS < (1 - 1/\sqrt{S})^2$, the left hand side of this equation is greater than $|\kappa_\nu - \gamma\kappa_\mu|$. The ratio on the right hand side is less then 1, so the right hand side is less than $|\kappa_\nu - \gamma\kappa_\mu|$. Consequently, this inequality is satisfied. Thus, at least in the large $\eta$ limit, all roots are stable.

## 4.3 Inference without periglomerular cells

In our model of sister mitral cells, periglomerular cells are critical to the inference process. This suggests a natural test of our model: remove them experimentally, see what happens to inference, and compare to the predictions of our model. Here we delineate those predictions.

For simplicity we'll assume only one odour is present, which, without loss of generality we take to be odour 1. The input, $y_i$, is, therefore, given by $y_i = A_{i1}x_1^*$. Setting $\mu_i^s$ to zero in Eq (33a) and eliminating $\lambda_i^s$ and $\nu_j$ in Eq (33), we see, after a small amount of algebra, that at equilibrium $x_j$ obeys

$$x_j = \frac{1}{\gamma}\left[\frac{1}{\sigma^2}\sum_i A_{ij}A_{i1}x_1^* - \frac{S}{\sigma^2}\sum_{s,i,k}W_{ij}^s W_{ik}^s x_k - \beta\right]^+. \tag{84}$$

We'll make the Ansatz that $x_1 \approx x_1^*$ and all the other $x_j$ are significantly smaller. This allows us to solve for $x_1$ by considering only $x_k = x_1$ on the right hand side of Eq (84). This still leaves us with a matrix equation. However, given the discussion in Sec. 4.2, to lowest order we can approximate the matrices in Eq (84) as the identity,

$$\sum_i A_{ij}A_{i1} \approx \delta_{j1} \tag{85a}$$

$$\sum_{s,i} W_{ij}^s W_{ik}^s \approx \delta_{jk}. \tag{85b}$$

With these approximations, the equation for $x_1$ becomes

$$x_1 \approx \frac{1}{\gamma}\left[\frac{1}{\sigma^2}\left(x_1^* - Sx_1\right) - \beta\right]^+, \tag{86}$$

which has the solution

$$x_1 \approx \frac{x_1^* - \beta\sigma^2}{S + \gamma\sigma^2}. \tag{87}$$

To recover the no leak case ($\varepsilon = 0$), we can set $S = 1$, because in that case Eq (84) corresponds to the MAP solution. Thus, the first observation is that eliminating the periglomerular cells reduces the inferred amplitude of the odour component present by a factor equal to the number of periglomerular cells (in the small noise—meaning small $\sigma^2$—limit).

We can also determine the effect on the incorrect odours. For $j \neq 1$, Eq (84) may be written

$$x_j \approx \frac{1}{\gamma}\left[\frac{1}{\sigma^2}\sum_i\left(A_{ij}A_{i1}x_1^* - S\sum_s W_{ij}^s W_{i1}^s x_1\right) - \frac{Sx_j}{\sigma^2} - \beta\right]^+ \tag{88}$$

where it is assumed that $j \neq 1$. On average, $W_{ij}^s = A_{ij}/S$. Consequently, when $j \neq 1$, on average $W_{ij}^s W_{i1}^s = A_{ij}A_{1i}/S^2$. Using this, and also using Eq (87) for $x_1$, this equation becomes

$$x_j \approx \frac{1}{\gamma}\left[\frac{(S + \gamma\sigma^2 - 1)x_1^* + \beta\sigma^2}{\sigma^2(S + \gamma\sigma^2)}\sum_{i=1}^M A_{ij}A_{i1} - \frac{Sx_j}{\sigma^2} - \beta\right]^+. \tag{89}$$

Because the elements of $A_{ij}$ have variance $1/M$, the sum over $i$ is a random variable with variance $1/M$. We thus have

$$x_j \approx \frac{1}{\gamma}\left[\frac{(S + \gamma\sigma^2 - 1)x_1^* + \beta\sigma^2}{\sigma^2(S + \gamma\sigma^2)}\frac{\xi_j}{\sqrt{M}} - \frac{Sx_j}{\sigma^2} - \beta\right]^+ \tag{90}$$

where $\xi_j \sim \mathcal{N}(0, 1)$. (Note that this underestimates the variance, because we are ignoring the additional variability in the matrix $W_{ij}^s W_{i1}^s$, so we'll be underestimating the effect of eliminating the periglomerular cells). This has the solution

$$x_j \approx \left[\frac{(S + \gamma\sigma^2 - 1)x_1^* + \beta\sigma^2}{(S + \gamma\sigma^2)^2}\frac{\xi_j}{\sqrt{M}} - \frac{\beta\sigma^2}{S + \gamma\sigma^2}\right]^+. \tag{91}$$

The main observation is that in the small noise regime, the regime we consider here, there's a big difference between no leak ($S = 1$) and infinite leak: in the former case, $x_j/x_1^* \sim \sigma^2/\sqrt{M}$; in the latter it scales as $1/S\sqrt{M}$. Because $\sigma^2 \ll 1/S$, this is a large effect.

This analysis suggests that eliminating periglomerular cells decreases the amplitude of the correctly inferred odours and increases the amplitude of the incorrect inferred odours, justifying our claim in the Discussion that eliminating periglomerular cells from the circuit would result in low amplitude, distributed activity.

## 4.4 Simulations

The base values of all parameters used in simulations are listed in Table 1. Membrane time constants for mitral and granule cells were set to be similar to the corresponding charging time constants $\tau_0$ in [41]. For simplicity the time constant for the periglomerular cells was set to be the same as that of the granule cells, and is consistent with [28]. To model each granule cell connecting to a single sister cell from each glomerulus, we selected for each glomerulus $i$ and granule cell $j$ a random sister cell; see Eq (18). The non-zero concentrations of the presented odours were set to 1, except in Figs 5, 6, 9 and 12, where the true odour had the default number $n = 3$ non-zero components but at concentrations of 0.8, 1, and 1.2, to aid visual assessment of convergence to the MAP solution. All simulations were initialized with zero activity in all cell populations.

To assess the variability of the various response characteristics we usually chose to present the same odour to different random instances of the olfactory bulb, rather than picking

different odours within the same olfactory bulb. Thus all references to trials are to the same odour presented to different olfactory bulbs unless stated otherwise.

In Fig 7 the Marchenko-Pastur parameter $v$ used to compute $\xi^{\text{high}}$ and $\xi^{\text{low}}$ was set to max $(1, n/M)$ i.e. to 1 in panel A, and to 55/50 in panel B.

The quantities in Fig 8 were computed as follows. Amplitude spectra for panel A and panel B were computed as the absolute value of the Fourier transform of the mitral cell responses in the time interval $t = 0.3 - 0.6$ seconds, averaged over all mitral cells and 20 trials. The decay time constants in panel C were computed from the slope of a linear fit to the log RMS error of the mitral cell activations relative to their final value, for the interval $t = 0.4 - 0.6$ seconds following odour onset, averaged over 20 trials.

To produce Fig 13 we simulated the response of one olfactory bulb to 100 random odours. Each glomerulus contained $S = 25$ sister cells; all other parameters were the same as in Table 1. Simulations were run for 2.1 seconds to allow the bulb to converge.

To generate cell-specific nonlinearities to model transformations from our model variables, $\lambda_i^s$, to those that might be experimentally recorded, $\tilde{\lambda}_i^s$, we first discretized the interval $[-6, 6]$, spanning the range of $\lambda_i^s$ values that we observed in our simulations, into 101 points $x_1, \ldots x_{101}$. We generated a covariance kernel $\Sigma$ on these points according to $\Sigma_{ij} = \exp(-(x_i - x_j)^2/8)$. We then generated $MS$ random samples (one for each of the $S$ sisters from $M$ glomeruli) from a 101-dimensional multivariate Gaussian with mean 0 and covariance $\Sigma$, yielding, for each sample, a function $y$ of $x$ with discrete domain. To render these functions monotonically increasing, we first computed the minimum slope along each. When the minimum was negative, we added a term linear in $x$ whose positive slope was 1.02 times larger than the absolute value of the minimum negative slope. This rendered the minimal slopes of the resulting discretized functions positive and the functions themselves monotonically increasing. The functions were then scaled to have the same $y$ range as $x$ range, and vertically offset so that their minimum y-value was zero. We used linear interpolation to expand the domain of these functions from the 101 discretization points to the full $[-6, 6]$ interval.

For all simulations we used the forward Euler method with a time step of $10^{-3}$ ms. To confirm that our networks performed MAP inference, we compared solutions to those found by the convex optimization package CVXPY [42] using the splitting conic solver (SCS), with `eps` set to $5 \times 10^{-13}$, applied to the MAP problem in Eq (7) expressed as the constrained optimization

$$\min_{\substack{\mathbf{x} \in \mathbb{R}^N \\ \mathbf{r} \in \mathbb{R}^M}} \sum_{j=1}^{N} \phi(x_j) + \frac{1}{2\sigma^2} \sum_{i=1}^{M} r_i^2 \quad \text{such that} \quad r_i = y_i - \sum_j A_{ij} x_j. \tag{92}$$

The code used to run the simulations and produce the figures are available at https://github.com/stootoon/sister-mcs-release.

## 4.5 Cortical readout

In our model, the concentrations of the odour components are stored in granule cells, which don't project outside of the olfactory bulb, and in fact lack axons entirely [43]. Thus, the granule cells can't communicate any information to the rest of the brain. This can be remedied by projecting mitral cell activity to cortical readouts $x_j^c$ via projection weights $U_{ij}^s$

$$\tau_c \frac{dx_j^c}{dt} = -\frac{\partial \phi(x_j^c)}{\partial x_j^c} + \sum_{i,s} U_{ij}^s \lambda_i^s. \tag{93}$$

In this circuit each cortical neuron $x_j^c$ is excited by the sister cells in the same way as the granule

cells in the bulb, but is not required to provide feedback to the bulb. When computation in the bulb converges, we have $\lambda_i^s = (y_i - \sum_j A_{ij} x_j)/\sigma^2$ (see Eq (33a) and recall that sister mitral cells all converge to the same activity), so that

$$\frac{\partial \phi(x_j^c)}{\partial x_j^c} = \frac{1}{\sigma^2} \sum_i \left( \sum_s U_{ij}^s \right) \left( y_i - \sum_j A_{ij} x_j \right). \tag{94}$$

Thus as long as the projection weights to the cortex satisfy

$$\sum_s U_{ij}^s = A_{ij}, \tag{95}$$

(analogous to Eq (14)) then cortical neuron $x_j^c$ will have the same fixed point as granule cell $x_j$. This means the output of the computation in the bulb can be read out in the cortex via a 1-to-1 correspondence between granule cells and cortical neurons. Thus basic olfactory inference can be performed entirely within the bulb, with the concomitant increase in computational speed, and the results can be read out in the cortex. As cortical feedback to the bulb, in particular to the granule cells, does in fact exist [43], its role may be to incorporate higher level cognitive information and task contingencies into the inference. We leave the exploration of these ideas to future work.

## 4.6 Application to other models

Our model used sister mitral cells to sparsify connectivity in a circuit performing inference under a linear model of olfactory transduction with Gaussian noise. Our approach, however, is quite general, and can be applied to more complex models. For example, in [2] the authors also consider a linear model of olfactory transduction, but with Poisson noise and a spike-and-slab prior on odour concentrations. Eqs (28) and (29) from their model translate, with minor redefinitions to be consistent with our notation, and minor simplifications to reduce clutter, to

$$\tau_\lambda \frac{d\lambda_i}{dt} = -\bar{r}_i \lambda_i^2 + r_i - \lambda_i \sum_k A_{ik}^{\mathrm{mg}} x_k^i \tag{96a}$$

$$\tau_x \frac{dx_k}{dt} = -x_k + \sum_j C_{kj} F_j(x_j^c) \tag{96b}$$

$$\tau_x \frac{dx_k^i}{dt} = -x_k^i + x_k A_{ki}^{\mathrm{gm}} \lambda_i \tag{96c}$$

$$\tau_c \frac{dx_j^c}{dt} = -x_j^c + \beta_j \left( \alpha_0 + F_j(x_j^c) \sum_i \lambda_i^2 A_{ij} \right) \tag{96d}$$

where the connectivity matrices $A_{ik}^{\mathrm{mg}}$, $A_{ki}^{\mathrm{gm}}$ and $C_{jk}$ are related to $A_{ij}$ by

$$A_{ij} = \sum_k A_{ik}^{\mathrm{mg}} A_{ki}^{\mathrm{gm}} C_{kj}. \tag{97}$$

There are several differences between this model and ours. First, the input, which in our model was $y_i$, is now stochastic: $r_i$ is the number of spikes in a bin of size $\Delta t$ generated from a Poisson process with rate proportional to $y_i$, and $\bar{r}_i$ is the expected number of spikes. Second, the granule cells and mitral cells communicate via dendro-dendritic connections at "spines", denoted $x_k^i$; this results in several connection matrices rather than just one. Third, the cortical

readout, $x_j^c$, feeds back to the olfactory bulb. Fourth, the nonlinearity, $F(x_j^c)$, which is defined in terms of the digamma function, is very different from ours. And fifth, the equation for the mitral cells has a term $\lambda_i^2$ on the right hand side.

What they have in common is that the connectivity matrices, $A_{ik}^{\mathrm{mg}}$ and $A_{ki}^{\mathrm{mg}}$, are dense, and so would require mitral cells to interact with nearly all granule cells. This results in the same all-to-all connectivity problem that we highlighted in Eq (10). But it can again be fixed using sister mitral cells and periglomerular cells,

$$\tau_\lambda \frac{d\lambda_i^s}{dt} = -(\lambda_i^s)^2 + y_i - \lambda_i^s S \sum_k W_{ik}^{\mathrm{mg},s} x_k^i - S\mu_i^s \tag{98a}$$

$$\tau_x \frac{dx_k^i}{dt} = -x_k^i + x_k \sum_s W_{ki}^{\mathrm{gm},s} \lambda_i^s \tag{98b}$$

$$\tau_\mu \frac{d\mu_i^s}{dt} = \lambda_i^s - \frac{1}{S}\sum_s \lambda_i^s \tag{98c}$$

$$\tau_c \frac{dx_j^c}{dt} = -x_j^c + \beta_j \left( \alpha_0 + F_j(x_j^c) \sum_i \sum_s (\lambda_i^s)^2 W_{ij}^s \right). \tag{98d}$$

Because of Eq (98c), at equilibrium all the sister mitral cells (all the $\lambda_i^s$) have the same value. Then, assuming, as above, that at $t = 0$ the average periglomerular activity is zero, it's easy to see that the sister mitral cells have the same equilibrium values as they do in Eq (96) if

$$\sum_s W_{ik}^{\mathrm{mg},s} = A_{ik}^{\mathrm{mg}} \tag{99a}$$

$$\sum_s W_{ki}^{\mathrm{gm},s} = A_{ki}^{gm} \tag{99b}$$

$$\sum_s W_{ij}^s = A_{ij}. \tag{99c}$$

Thus, sister mitral cells can be used in more complicated models than the purely linear one we considered here.

## Acknowledgments

We thank members of the Gatsby Unit and the Latham and Schaefer labs for useful discussions. For the purpose of Open Access, the authors have applied a CC BY public copyright licence to any Author Accepted Manuscript version arising from this submission.

## Author Contributions

**Conceptualization:** Sina Tootoonian, Peter E. Latham.

**Data curation:** Sina Tootoonian.

**Formal analysis:** Sina Tootoonian, Peter E. Latham.

**Funding acquisition:** Andreas T. Schaefer, Peter E. Latham.

**Investigation:** Sina Tootoonian, Peter E. Latham.

**Methodology:** Sina Tootoonian, Peter E. Latham.

**Resources:** Andreas T. Schaefer, Peter E. Latham.

**Software:** Sina Tootoonian.

**Validation:** Sina Tootoonian, Peter E. Latham.

**Visualization:** Sina Tootoonian.

**Writing – original draft:** Sina Tootoonian, Andreas T. Schaefer, Peter E. Latham.

**Writing – review & editing:** Sina Tootoonian, Andreas T. Schaefer, Peter E. Latham.

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
