## [Decision Letter · Decision Letter 0]

1 Sep 2021

Dear Dr. Tootoonian,

Thank you very much for submitting your manuscript "Sparse connectivity for MAP inference in linear models using sister mitral cells" for consideration at PLOS Computational Biology.

As with all papers reviewed by the journal, your manuscript was reviewed by members of the editorial board and by several independent reviewers. In light of the reviews (below this email), we would like to invite the resubmission of a significantly-revised version that takes into account the reviewers' comments.

As you can see both reviewers valued the modelling idea but raised concerns about a potential departure from available biological evidence and an apparent lack of performance for specific scenarios. Reviewer #1 makes concrete suggestions of how to provide more compelling evidence for the introduction of mitral and sister cells into the model. It would be great if you addressed these first two points as the two key major points of reviewer #1.

We cannot make any decision about publication until we have seen the revised manuscript and your response to the reviewers' comments. Your revised manuscript is also likely to be sent to reviewers for further evaluation.

Sincerely,

Stefan Kiebel

Associate Editor

PLOS Computational Biology

Samuel Gershman

Deputy Editor

PLOS Computational Biology

Reviewer's Responses to Questions

**Comments to the Authors:**

Reviewer #1: See the attached document.

Reviewer #2: Tootoonian and colleagues developed a rigorous model for sensory inference aimed at obtaining a maximum a posteriori (MAP) inference solution. Specifically, they are taking advantage of the olfactory circuit architecture to find a solution supported by biologically plausible sparse connectivity. This is in contrast with standard MAP implementations that require dense, biologically unrealistic, all-to-all connectivity. The authors’ model assigns different sister mitral cells which receive input from the same common glomerulus the job of extracting different latent variables and posits that each sister cell connects reciprocally and symmetrically to different sets of inhibitory granule cells interneurons. This strategy enables each mitral cell to interact with a smaller number of granule cells (by a factor S, ~25 given the number of sister cells per glomerulus). They further achieve coordination among sister mitral cells, necessary for their model implementation to arrive to MAP solution, by engaging back propagating action potentials from the sister mitral cells to drive the firing of periglomerular cells which, in turn, inhibit the sister mitral cells and drive them to converge at equilibrium to the same activity patterns, or at the same ratio in an odor independent fashion (a strong prediction of the model). In the current implementation, the readout units are the granule cells, which nevertheless do not relay information directly to the rest of the brain. The authors argue that a similar architecture with matched weights can be implemented at the level of pyramidal cells in the cortex that can also extract the same kind of information w/o having to relay it back to the mitral cells. The model solves the inference problem successfully for a small number of odors in the sensory scene, but its performance becomes poorer and poorer with increasing number of odors within the range sampled (up to 10). In my opinion, model is elegant and clearly spelled out, the manuscript is well-written, with numerous explanations of the assumptions and possible caveats and solution, and, thus, a pleasure to read. I also found the idea of using the sister cells as means to implement the necessary sparse connectivity very clever. However, my enthusiasm was reduced by the poor performance of the model in conditions where the number of odorants to represent is relatively small (>3-4), as well as by several implementation choices and its predictions that, in my opinion, depart from known biological evidence. I list below my main concerns:

1. No detail/reasoning is provided with respect with how the sparse prior is stored in the brain/model architecture. How is the prior information represented, stored and how is it acquired?

2. The model does not seem to work efficiently for simple sensory scenes where multiple odorants are present. It appears that starting from n=3 odors, the model starts to recover many more odors than are actually present.

3. The hard prediction of the model that all sister cells converge to the same activity is not necessarily supported by previous experimental work in anaesthetized and awake mice (Dhawale et al., 2010, Arneodo et al., 2017) that identified sister mitral cells in vivo and showed that their activity patterns are non-redundant and, if anything, diverge in time. Average firing rate changes of sister cells tend to be indeed somewhat similar to each other (more so than pairs of non-sister cells), but they span a wide range. Furthermore, the ratio of their responses appears to not odor-independent, but to vary as function of the odor presented. Their average odor responses at steady state are not simply scaled versions of each other, displaying instead a wide distribution of response correlations. Throughout the manuscript, the predictions of the model with respect to sister mitral cells converging to same activity patterns at equilibrium and whether they have or not the same ratio of response to different odors vary across different sections. With respect to their spike-timing, experimental work found that sister cells become decorrelated in an odor specific manner, both with respect to each other, as well as with the respiration clock. The model does not address these biological results (in the current implementation, it does not discuss the transmission of information with respect to different phases of the respiration cycle).

4. The authors choose to not implement cortical feedback in this version of their model (though in previous work they have, Grabska-Barwinska et al., 2017), which constraints its relation to the biology, since a cortical feedback is a key feature of the vertebrate olfactory circuit. For example, firing of the granule cells is heavily dependent of cortical input and controls the timing and gain of mitral cell firing. Also, cortical feedback may be important in relaying the required sparse prior.

5. As the authors note, the outputs of individual glomeruli consist of both mitral and tufted and sister cells with very different intrinsic properties (including substantially different firing properties), local wiring and projection patterns. However, in the current implementation both populations are bundled together. This is to some degree a technical detail, but it does have important consequences on key requirements/benefits of the model: whether it is realistic to consider convergence to similar firing patterns in time (this appears different for tufted and mitral cells) and the size of S (Schwarz et al., 2018).

6. The choice of periglomerular cells activated by back-propagating action potentials of mitral cells as a source to regulate the similarity of sister cells is interesting, but further discussion would be needed in terms of the assumptions made with respect to connectivity within the glomerulus of these cells and their timing of spiking with respect to the firing of mitral cells. The activity profiles of mitral, periglomerular and granule cells shown in Figs. 9 & 12 appear quite different from documented activity patterns of such cells in the brain and converge to a stable steady state within 100-200 ms from stimulus onset.

**Have the authors made all data and (if applicable) computational code underlying the findings in their manuscript fully available?**

Reviewer #1: Yes

Reviewer #2: Yes

PLOS authors have the option to publish the peer review history of their article (what does this mean?). If published, this will include your full peer review and any attached files.

Reviewer #1: **Yes: **Darío Cuevas Rivera

Reviewer #2: **Yes: **Dinu F Albeanu
---

## [Decision Letter · Decision Letter 1]

9 Dec 2021

Dear Dr. Tootoonian,

Thank you very much for submitting your manuscript "Sparse connectivity for MAP inference in linear models using sister mitral cells" for consideration at PLOS Computational Biology. As with all papers reviewed by the journal, your manuscript was reviewed by members of the editorial board and by several independent reviewers. The reviewers appreciated the attention to an important topic. Based on the reviews, we are likely to accept this manuscript for publication, providing that you modify the manuscript according to the review recommendations.

Sincerely,

Stefan Kiebel

Associate Editor

PLOS Computational Biology

Samuel Gershman

Deputy Editor

PLOS Computational Biology

[LINK]

Reviewer's Responses to Questions

**Comments to the Authors:**

Reviewer #1: The authors have addressed most of the major and minor points from the previous round. However, one important point remains unaddressed.

The transition from equation 9 to 10 seems to be a simple change of variables, where \\lambda is defined as \\lambda_i = \\frac{1}{\\sigma^2}(y_i - \\sumA_{ik}x_k). Note that \\frac{1}{\\sigma^2} has to be a part of the definition for equation 10b to hold, contrary to the text description of “factor out the term in parenthesis”.

Given the definition of \\lambda, equation 10a can be rewritten as:

\\tau_\\lambda \\frac{d\\lambda_i}{dt} = -\\lambda_i + \\frac{1}{\\sigma^2}(y_i - \\sumA_{ik}x_k) = -\\lambda_i + lambda_i = 0

This means that \\lambda_i is fixed in time and the results that follow are based on the false premise that \\lambda and (x, y) are independent from each other.

As it stands, the description of the model up to equation 10 is written as if it were a derivation (and the word is used throughout the manuscript). However, given the point above, perhaps the authors meant to introduce an arbitrary model (eqn 10) and then show that it performs MAP, just as the system in eqn 9 does.

While it might sound like a distinction without a difference, it is important to note that equations 9 and 10 do not describe the same system at all, but do converge to the same equilibrium point. If this was the authors’ intention, please indicate so in the manuscript, especially in the transition from equation 9 to 10.

Minor comment:

For completeness, a condition of continuity could be added to the transformations f, such that invertible implies monotonic (as said in line 435).

Reviewer #2: The authors have addressed several of my concerns and substantially improved the manuscript which, I think, will have an important impact in the field. I could still quibble with the exact implementation schemes (no cortical feedback) and the convergence of sister mitral cells (happy to share the data on their odor response spectra), but, in my opinion, these points could be addressed in further manuscripts, and this one is now ready for publication.

**Have the authors made all data and (if applicable) computational code underlying the findings in their manuscript fully available?**

Reviewer #1: Yes

Reviewer #2: Yes

PLOS authors have the option to publish the peer review history of their article (what does this mean?). If published, this will include your full peer review and any attached files.

Reviewer #1: **Yes: **Dario Cuevas Rivera

Reviewer #2: **Yes: **Dinu Albeanu

Figure Files:

Data Requirements:

Reproducibility:

References:

---

## [Editor Report · Decision Letter 2]

5 Jan 2022

Dear Dr. Tootoonian,

We are pleased to inform you that your manuscript 'Sparse connectivity for MAP inference in linear models using sister mitral cells' has been provisionally accepted for publication in PLOS Computational Biology.

Best regards,

Stefan Kiebel

Associate Editor

PLOS Computational Biology

Samuel Gershman

Deputy Editor

PLOS Computational Biology

---

## [Editor Report · Acceptance letter]

25 Jan 2022

PCOMPBIOL-D-21-01274R2

Sparse connectivity for MAP inference in linear models using sister mitral cells

Dear Dr Tootoonian,

I am pleased to inform you that your manuscript has been formally accepted for publication in PLOS Computational Biology. Your manuscript is now with our production department and you will be notified of the publication date in due course.

With kind regards,

Livia Horvath
